# Donor regulatory T cells rapidly adapt to recipient tissues to control murine acute graft-versus-host disease

David J. Dittmar [1,3,5], Franziska Pielmeier[1,5], Nicholas Strieder [2], Alexander Fischer [1], Michael Herbst [1,4], Hanna Stanewsky[1], Niklas Wenzl[2], Eveline Röseler[2], Rüdiger Eder[1], Claudia Gebhard[2], Lucia Schwarzfischer-Pfeilschifter[1], Christin Albrecht[1], Wolfgang Herr[1], Matthias Edinger [1,2,6] ✉, Petra Hoffmann [1,2,6] ✉ & Michael Rehli [1,2,6] ✉

The adoptive transfer of regulatory T cells is a promising strategy to prevent graft-versus-host disease after allogeneic bone marrow transplantation. Here, we use a major histocompatibility complex-mismatched mouse model to follow the fate of in vitro expanded donor regulatory T cells upon migration to target organs. Employing comprehensive gene expression and repertoire profiling, we show that they retain their suppressive function and plasticity after transfer. Upon entering non-lymphoid tissues, donor regulatory T cells acquire organ-specific gene expression profiles resembling tissue-resident cells and activate hallmark suppressive and cytotoxic pathways, most evidently in the colon, when co-transplanted with graft-versus-host disease-inducing conventional T cells. Dominant T cell receptor clonotypes overlap between organs and across recipients and their relative abundance correlates with protection efficacy. Thus, this study reveals donor regulatory T cell selection and adaptation mechanisms in target organs and highlights protective features of Treg to guide the development of improved graft-versus-host disease prevention strategies.

Allogeneic hematopoietic stem cell transplantation (alloHSCT) is a curative treatment option for many patients with hematologic malignancies but bears the risk of inducing graft-versus-host-disease (GvHD), a donor T cell-mediated immune reaction damaging host tissues. Particularly in its steroid-refractory acute form, GvHD remains the leading cause of non-relapse morbidity and mortality after alloHSCT[1,2] and thus warrants the development of new prevention and treatment strategies[3,4].

CD4+CD25+Foxp3+ regulatory T cells (Treg) play a key role in peripheral tolerance and protect from autoimmune diseases[5–7]. Recent studies revealed a substantial heterogeneity among Treg that

originates from different sites of induction (thymus versus periphery, especially the GI tract), different developmental states (naïve, tissue precursor, central memory or effector) as well as different localization (lymphoid versus non-lymphoid organs)[8–11]. Thus, immune homeostasis depends not only on a sufficient Treg pool size but also on a balanced composition of the Treg population.

Thymus-derived donor Treg co-transferred with the stem cell graft can protect from acute GvHD (aGvHD) without impairing the beneficial graft-versus-leukemia (GvL) effect, as demonstrated in murine models and first clinical trials[12–16]. Initial homing of donor Treg to secondary lymphoid organs is required for optimal GvHD

[1]Department of Internal Medicine III, University Hospital Regensburg, 93053 Regensburg, Germany. [2]Leibniz Institute for Immunotherapy, 93053 Regensburg, Germany. [3]Present address: BioNTech SE, 82061 Neuried, Germany. [4]Present address: Institute of Experimental Immunology, Research Unit Tumorimmunology, University of Zurich, Zurich, Switzerland. [5]These authors contributed equally: David J. Dittmar, Franziska Pielmeier. [6]These authors jointly supervised this work: Matthias Edinger, Petra Hoffmann, Michael Rehli. ✉e-mail: matthias.edinger@ukr.de; petra.hoffmann@ukr.de; michael.rehli@ukr.de

prevention[17,18]. This is followed by their expansion and migration to non-lymphoid GvHD target organs (such as colon, liver and skin) and the local suppression of tissue-destructive allo-responses[19,20]. Despite this knowledge on early donor Treg migration, little is known about the selection and adaptation processes in this phase, in particular upon relocation to peripheral non-lymphoid tissues.

In most studies performed to date donor Treg were used directly after ex vivo enrichment. However, for future applications, including therapeutic intervention in progressive acute or chronic GvHD or treatment of autoimmune disorders, higher cell numbers might be required and thus in vitro expansion of Treg prior to injection might be necessary. We have previously established protocols for the efficient polyclonal expansion of GMP-grade human Treg[21,22] and presently investigate such cell products in phase I/II clinical trials (EudraCT #2012-002685-12 and 2016-003947-12). In addition, we were able to prove the efficacy of polyclonally expanded Treg in a murine model of aGvHD[23]. Yet, some previous publications suggested a superior GvHD-protective effect of donor-Treg if they were primed against host-APCs during the in vitro expansion phase[16,24,25]. We therefore compared the clinical efficacy of allo-antigen primed vs. polyclonally expanded donor Treg and comparatively examined the selection and organ-specific adaptation of the cell products in a complete MHC-mismatched BMT model. For this purpose, we re-isolated donor Treg and conventional T cells (Tconv) from spleen, liver and colon 7 days after BMT, analyzed the cells by multiparametric flow cytometry and performed bulk and single-cell transcriptome as well as TCR repertoire profiling. Our results show that cultured Treg maintain their functionality even after

extensive in vitro expansion. Upon migration to non-lymphoid host organs, in particular the colon, donor Treg rapidly acquired organ-specific gene expression profiles resembling those of their tissue-resident counterparts, while simultaneously upregulating a broad range of suppression programs. The distribution of dominant clonotypes across different organs and even across different recipients indicates that early migration and GvHD protection in the MHC-disparate setting is preferentially driven by ubiquitously expressed allo-antigens. Overall, this comprehensive atlas on clonal distribution and gene expression reveals for the first time the plasticity and site-specific functional adaption of adoptively transferred donor Treg after alloBMT.

## Results

### Alloantigen-specific and polyclonal donor Treg expansion protocols generate cell products with similar phenotype but different TCR repertoire diversity

We previously demonstrated that the therapeutic transfer of in vitro expanded murine donor Treg ameliorates clinical and histologic signs of aGvHD and significantly improves survival after MHC-mismatched BMT[23]. To further explore the specificities and functional properties of transplanted donor Tregs, we now employed an aGvHD prophylaxis model and compared two different in vitro expansion strategies: polyclonal expansion (poly) using CD3/28 beads, and allo-antigen specific expansion (allo) through Treg culture in the presence of recipient-type antigen-presenting cells (APC), as schematically outlined in Fig. 1a.

As expected, poly and alloTreg cultures differed in their in vitro expansion kinetics, with allo-specific expansion being delayed as

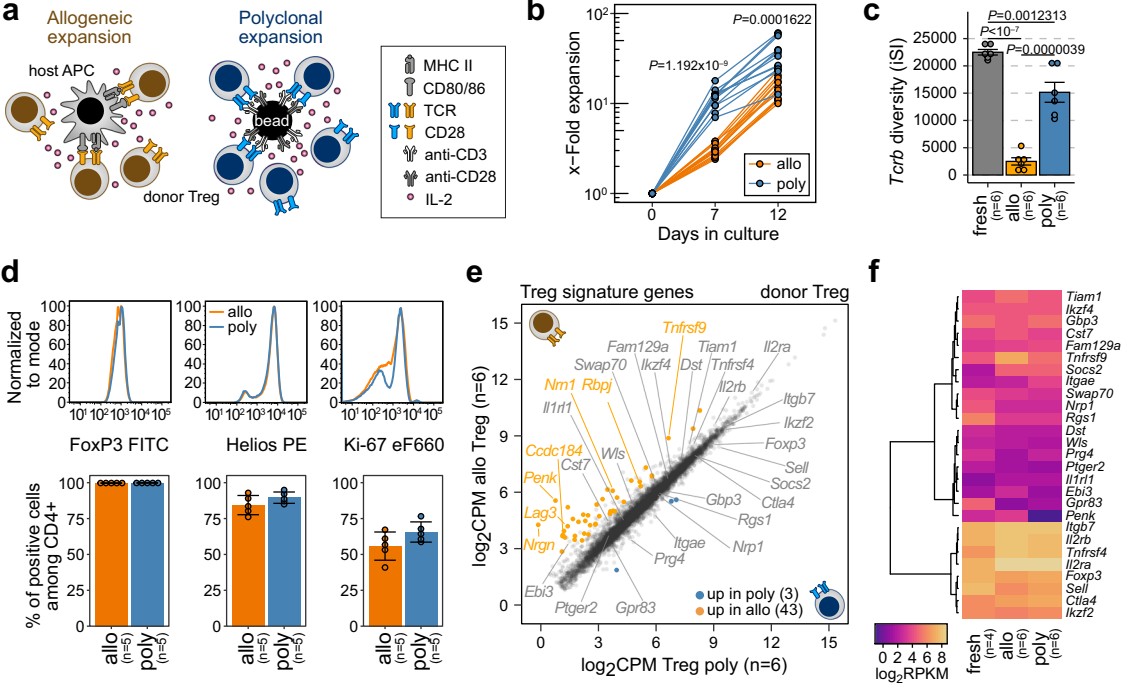

**Fig. 1 | Polyclonal versus alloantigen-specific expansion of donor Treg generates cell products with similar phenotypic properties but different TCR diversities. a** Schematic presentation of in vitro expansion strategies. **b** In vitro expansion rates of polyclonal (poly) and alloantigen-specific (allo) Treg cultures (n = 12 each). Two-tailed t-test. **c** Barplot of Tcrb clonotype diversities (iSI, inverse Simpson index) in independent isolations of C57BL/6 Treg from spleen (fresh, d0) or expanded Treg (allo, poly, d11–14) as measured by UMI-based sequencing of 5′RACE products of Tcrb mRNAs. Bars represent means ± SEM of n = 6 independent experiments, significant differences between groups are indicated above bars (one-way ANOVA with Tukey's post hoc test). Individual data points are shown as dots. **d** FACS analysis of the expression of canonical Treg markers FoxP3 and Helios, as well as Ki-67 as a marker for proliferation. Representative expression profiles of

polyclonal or alloantigen-specific in vitro expanded Treg (upper panels) and percent marker-positive cells in five independent Treg cultures (lower panels) are shown. Representative gating examples for each marker are shown in Supplementary Fig. 1e. Bars in the lower panels represent means ± SD of the indicated number of independent biological replicates. Individual data points are shown as dots. **e** Scatter plot indicating genes differentially expressed in allo versus polyTreg (as measured by RNA-seq, n = 6 independent experiments each). Treg-signature genes (based on[26] plus additional markers, listed in Supplementary Table 1) are labeled. **f** Heatmap comparing expression levels of Treg signature genes in independent isolations of C57BL/6 Treg from spleen (fresh, d0) or in vitro expanded Treg (allo, poly). **b**–**f** Source data are provided as a Source Data file.

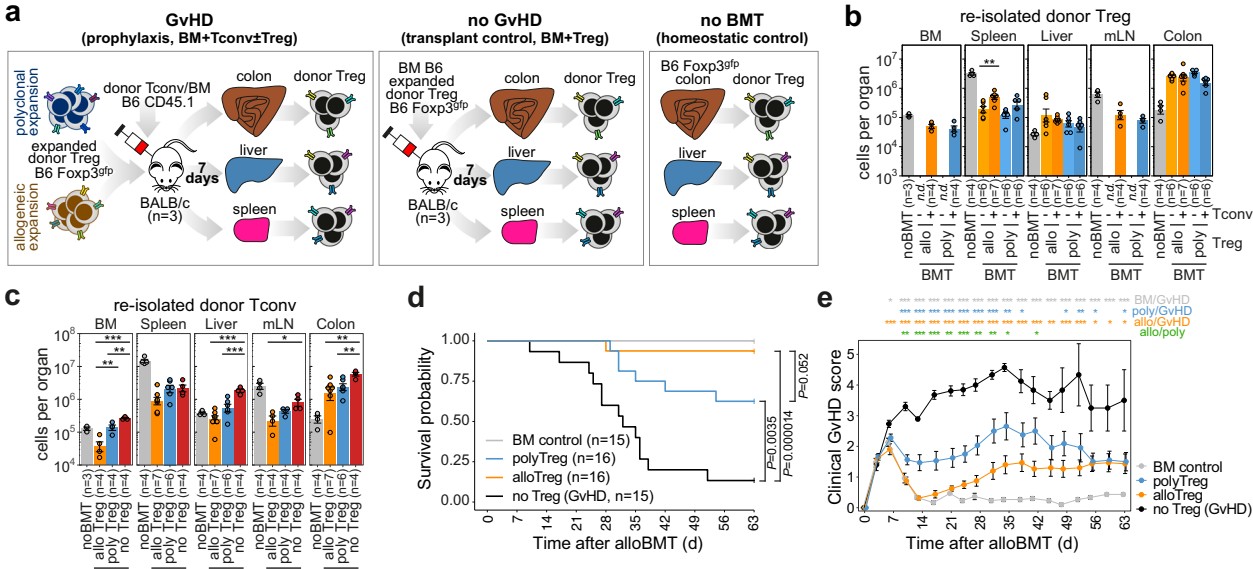

**Fig. 2 | Transplanted donor Treg migrate into GvHD target organs and alleviate transplant-associated complications in a murine MHC-disparate BMT model. a** Schematic overview of the experimental GvHD model and its controls. Left panel: GvHD in BALB/c recipients is induced by MACS-purified conventional CD4+CD25− donor T cells (Tconv; $1 \times 10^6$ cells/recipient) transferred together with T cell-depleted (TCD) BM ($2.5 \times 10^6$ cells/recipient), both from congenic CD45.1+ C57BL/6 mice. FACS-purified and in vitro expanded donor Treg from CD45.2+ Foxp3gfp mice are additionally given on d0 at a 1:1 ratio with Tconv for GvHD prevention (prophylaxis). Middle panel: the "no GvHD" transplantation control, which corresponds to the GvHD setting, but without transfer of Tconv. Right panel: organ resident C57BL/6 Treg serving as homeostatic control (no BMT). In both BMT settings, Treg (and Tconv) cells were re-isolated from the indicated organs 7 days after transfer for further analyses. **b** Bar plots showing absolute Treg cell numbers in the respective organs of C57BL/6 donor mice (noBMT) or absolute donor Treg cell numbers in transplanted BALB/c recipients that received allo or polyTreg together with TCD

BM (no GvHD) or TCD BM and additional Tconv (GVHD prophylaxis). **c** Bar plots showing absolute Tconv cell numbers in the respective organs of "noBMT" donor mice or absolute numbers of donor Tconv in recipients transplanted with TCD BM and Tconv in the presence of allo or polyTreg (prophylaxis) or without Treg (no Treg, GvHD). **b, c** Bars represent mean values ± SEM and dots represent data from individual recipients (numbers of independent recipient animals are indicated). Significant differences between BMT groups are indicated above bars. (One-way ANOVA with Tukey's post hoc test, except: BM and mLN in (**b**), two-tailed *t*-test; liver in (**b**), Kruskal–Wallis test with Dunn's post hoc test). **d** Survival of the indicated BMT recipient groups. Significant differences in survival are indicated on the right (Log rank test with Bonferroni correction). **e** Corresponding clinical GvHD scores for animals in (**d**). Data are presented as mean values ± SEM. Significant differences between treatment groups at individual data points are indicated on the top (one-way ANOVA with Tukey's post hoc test). **b–e** *p < 0.05, **p < 0.01, ***p < 0.001. **b–e** Source data and exact *p* values (**b, c, e**) are provided as a Source Data file.

---

compared to polyclonal expansion (Fig. 1b). The comparison of TCR α- and β-chain diversities between the starting populations (Treg isolated from spleen) and in vitro expanded cells showed that polyTreg largely retained their broad TCR repertoire, while allo-specific expansion resulted in a narrowing of the TCR repertoire (results for *Tcrb* are shown in Fig. 1c, for *Tcra* in Supplementary Fig. 1a). Otherwise, the two in vitro-expanded Treg cell products displayed a comparable expression of canonical Treg markers, such as Foxp3 and Helios and showed a similar proliferative potential, as determined by Ki-67 expression (Fig. 1d) as well as suppressive activity, as determined in T cell suppression assays (Supplementary Fig. 1b, c). Likewise, transcription profiling using RNA sequencing (RNA-seq) of in vitro expanded Treg at the day of BMT revealed only few differences between expansion protocols, with the large majority of Treg signature genes[26] being stably expressed in both, allo and polyTreg (Fig. 1e, f). Gene set enrichment analysis (GSEA) using Hallmark and Kegg pathway gene sets revealed a slight shift in the activity of pathways related to the culture environment (Glycolysis, Hypoxia, IL-2/STAT-5-Signaling) toward alloTreg, as shown in Supplementary Fig. 1d. Nevertheless, poly and alloTreg were generally overlapping in gene expression profiles, suggesting that the distinct expansion protocols had no major impact on pivotal signature genes and core Treg features.

### Donor-derived allo and polyTreg show similar migration patterns and rapidly accumulate in non-lymphoid organs after transfer into alloBMT recipients

To characterize the selection and adaptation processes of donor Treg in vivo, we comparatively examined polyclonally and alloantigen-

specifically expanded donor Treg populations after re-isolation from allogeneic hosts on d7 after transplantation. The treatment group consisted of alloBMT recipients that received T-cell-depleted bone marrow (TCD-BM) together with Tconv and Treg cells. In this setting, Treg actively suppress GvHD that would otherwise be initiated by Tconv (GvHD prophylaxis; BM+Tconv±Treg). Mice that received only TCD-BM and Tconv (but no Treg) represented the GvHD group (GvHD; BM+Tconv). One control group received only TCD-BM and Treg. In this setting GvHD is not expected due to the lack of Tconv and thus Treg dissemination and differentiation can be examined in alloBMT recipients in the absence of GvHD (no GvHD; BM+Treg). Untreated donor mice served as a second control group and source of steady-state organ-resident Treg (no BMT; see Fig. 2a for a schematic representation of experimental groups). As shown in Fig. 2b, transferred donor Treg were detected in higher than physiologic numbers in the non-lymphoid GvHD target organs colon and liver of alloBMT recipients, while fewer than normal Treg were detectable in recipient lymphoid organs such as spleen and mesenteric lymph nodes 7d after BMT (for relative proportion among leukocytes see Supplementary Fig. 2a, b). Interestingly, we observed neither significant differences in organ distribution between alloTreg and polyTreg nor in the expression of homing receptors such as CCR9, CD103 and LPAM-1 (Supplementary Fig. 2d). This suggests that pre-selection of donor Treg toward allo-antigens has little impact on their initial migratory behavior. In line with previous findings by us[14,23] and others[18,27], absolute cell numbers as well as relative proportions of donor Tconv in GvHD target organs were significantly reduced after donor Treg co-transfer (Fig. 2c and Supplementary Fig. 2c), with alloTreg being slightly more potent than

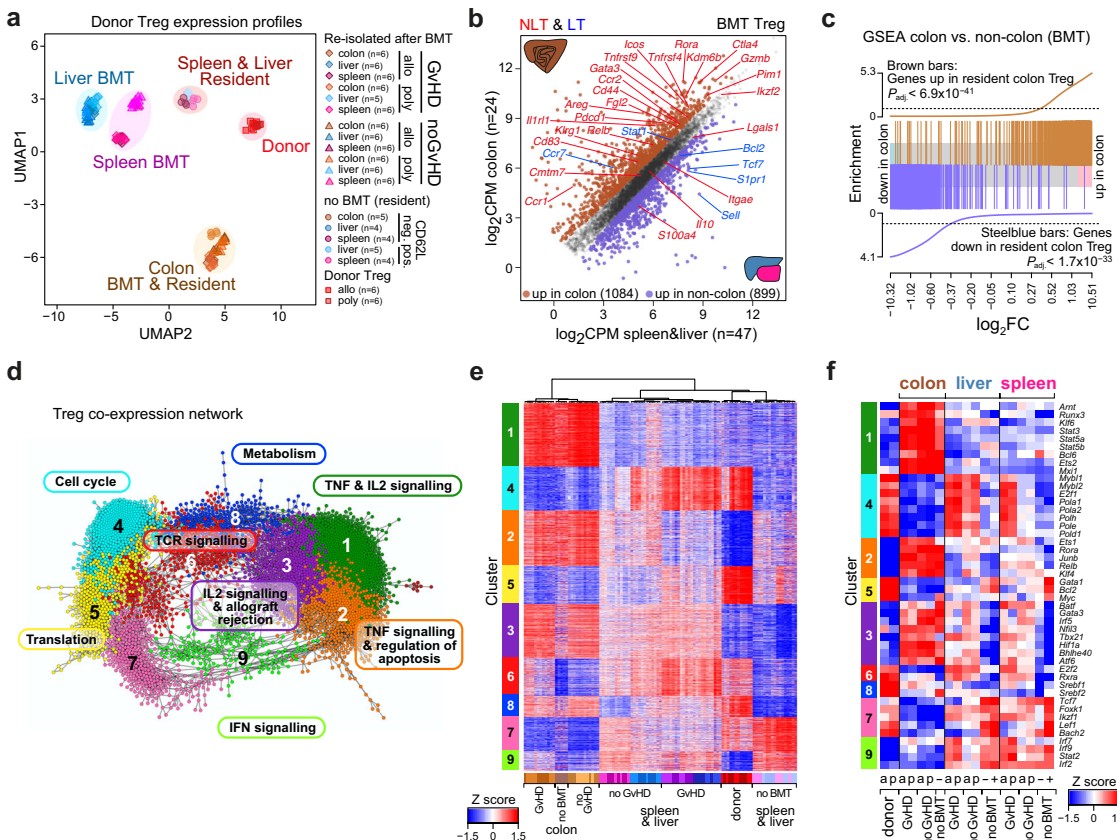

**Fig. 3 | Rapid, environment-dependent rewiring of Treg gene expression programs after BMT. a** UMAP embedding of donor Treg either after in vitro expansion before transfer ("Donor"), or re-isolated from recipient organs 7 days after transfer, as well as tissue-resident CD62L⁺ (naïve) and CD62L⁻ (effector/memory) Treg from non-transplanted animals, analyzed by RNA-seq and colored by their origin. Unlike spleen and liver, colon harbors only CD62L⁻ Treg. Data are from two independent experiments per BMT group (with $n = 3$ animals/experiment receiving the same graft). **b** Scatter plot of global gene expression (logCPM > 1, logRPKM > 1) comparing donor Treg (after BMT, with or w/o Tconv) re-isolated from colon versus non-colon samples. Differential genes are highlighted by coloring ($p_{adj.|FC|>1.5} > 0.05$, qlf-test). NLT and LT signature genes defined by Miragaia et al.[30] are highlighted. **c** Barcode plots presenting GSEA results for the indicated gene lists across the logFC ranking of genes based on the comparison shown in (**b**). Enrichment $p$ values of two-sided rotation gene set tests are given. **d** Main component of the gene co-expression network generated with Graphia ($r ≥ 0.845$, k-NN with $k = 4$, descending rank order, Louvain cluster granularity 0.6). For each co-expression cluster the top enriched functional annotation is given (except for cluster 7). More detailed functional annotation is provided in Supplementary Fig. 3m. **e** Heatmap presenting hierarchically clustered (sample-wise) and scaled expression data of genes included in the network shown in (**d**). Samples are color-coded as in (**a**). **f** Heatmap presenting averaged and scaled expression data of transcription factor genes associated with individual co-expression clusters. **b**, **c**, **e**, **f** Source data are provided as a Source Data file.

polyTreg, particularly in BM, mLN and liver. Likewise, both Treg populations convey significant protection from GvHD mortality (Fig. 2d), with alloTreg being more effective than polyTreg, probably due to the earlier and more profound inhibition of Tconv proliferation and the rapid amelioration of clinical manifestations (Fig. 2e). Nevertheless, these data clearly show that donor Treg, irrespective of their previous in vitro expansion protocol, accumulate in non-lymphoid GvHD target tissues 7 days after transfer, markedly suppress the expansion of co-transferred Tconv cells and efficiently protect recipients from lethal GvHD.

**Host tissues imprint transcription networks in transplanted donor Treg**

To further evaluate their phenotypic and functional characteristics upon transfer into allogeneic hosts, we performed comprehensive transcriptome analyses of donor Treg re-isolated from spleen, liver and colon of BALB/c recipients on d7 after transfer and compared the data sets to those from in vitro expanded Treg cell products before transfer ("Donor"). Since Treg cultures were set up with naïve (CD62L⁺) cells that retain CD62L expression during in vitro expansion but downregulate the marker after transfer into allogeneic recipients (Supplementary Fig. 2d), naïve and effector/memory (CD62L⁻) Treg

populations isolated from the respective organs of non-transplanted donor mice ("no BMT/resident") were also included in the analysis. Sorting strategies for organ-resident CD62L⁺ and CD62L⁻ Treg as well as transferred and re-isolated donor T cell subpopulations are outlined in Supplementary Fig. 3a, b. The clustering of gene expression data across all conditions and replicates grouped Treg according to their organ origin, illustrating that the organ microenvironment shapes the transcription profile of both, physiologically resident Treg and adoptively transferred donor Treg after alloBMT (Fig. 3a). In contrast, differences between re-isolated allo and polyTreg were subtle (Supplementary Fig. 3c) and overlapped with the minor differences already observed between in vitro cultured allo vs. poly donor Treg before transfer (compare Fig. 1e; a corresponding GSEA is shown in Supplementary Fig. 3d, examples for gene expression profiles are presented in Supplementary Fig. 3e). This indicates a "memory" for in vitro-induced differences for selected genes. In addition, GSEA of Hallmark and Kegg pathway gene sets revealed an enrichment of type-I IFN response genes among re-isolated polyTreg (Supplementary Fig. 3f). This was mainly evident in the spleen and suggests a slightly more inflammatory environment in this organ 7 days after polyTreg as compared to alloTreg application. Overall, however, re-isolated allo- and polyTreg showed no major differences. Due to their similarity and

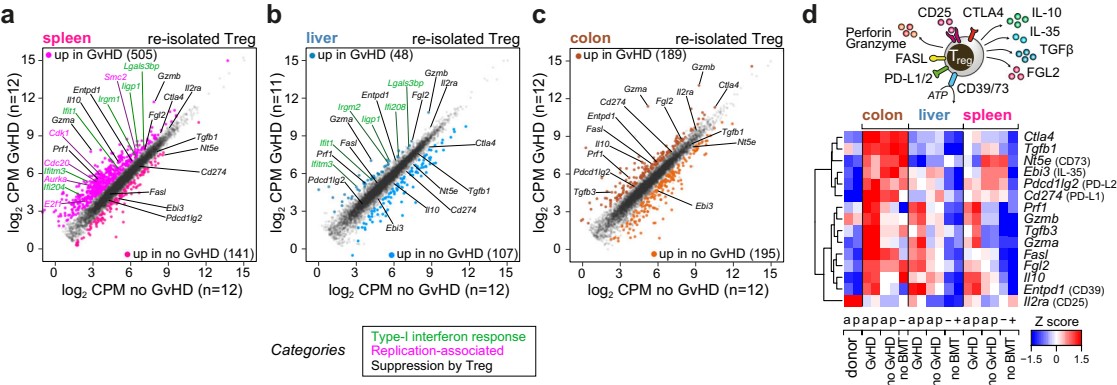

**Fig. 4 | Suppressive gene expression programs in organ-infiltrating donor Tregs upon GvHD induction. a–c** Scatter plots of global gene expression (logCPM > 1, logRPKM > 1) comparing donor Treg re-isolated from the indicated organs after BMT ± Tconv. Differential genes are highlighted categorically by coloring ($p_{\text{adj.|FC|>1.5}}$ > 0.05, qlf-test), as indicated below (**a, b**). **d** Heatmap presenting hierarchically clustered and scaled expression data of genes associated with suppressive Treg functions which are schematically depicted on top of the plot (a: allo, p: poly, +: CD62L⁺, −: CD62L⁻). **a–d** Source data are provided as a Source Data file.

to increase statistical power, we further on combined samples of differentially expanded Treg subpopulations for comparative gene expression analyses.

The UMAP clustering (Fig. 3a) illustrates that spleen and liver Treg samples were close yet distinct from each other, while colon Treg samples were most separate from all other samples. Hence, we first compared gene expression profiles of donor Treg re-isolated from colon to those retrieved from spleen and liver of alloBMT recipients. As shown in Fig. 3b, gene expression in colon-derived donor Treg was indeed very different from that of spleen- and liver-derived cells, with many genes significantly upregulated in either colon or spleen/liver (highlighted by coloring in Fig. 3b). Since only colon cells underwent enzymatic treatment at 37 °C during isolation, we cannot definitely exclude a contribution of differential stress responses upon Treg isolation from target tissues to differential gene expression. However, given the observed magnitude of differences between colon and non-colon Treg compared to the few genes known to be induced by enzymatic tissue dissociation procedures[28,29], the large majority of changes represent differential tissue imprints. In line with this, differentially expressed genes overlapped substantially with gene sets (labeled in Fig. 3b) previously classified in a single-cell analysis of tissue-resident Treg as non-lymphoid tissue (NLT)- and lymphoid tissue (LT)-specific, respectively[30]. A similar overlap was observed with gene sets distinguishing colon- from spleen- and liver-resident, memory/effector-type Treg (Fig. 3c and Supplementary Fig. 3g–k).

For colon-colonizing cells this is particularly remarkable, as re-isolated donor Treg represent thymus-derived cells, whereas colon-resident (no BMT) CD62L⁻ Treg are likely a mixture of thymus-derived and peripherally-induced (p)Treg, as indicated by their high expression of several pTreg markers[31], such as *Rorc, Maf, Gpr15* (Supplementary Fig. 3l) and *Il10* (Supplementary Fig. 4d). Thus, our data demonstrate the dominant imprint of the colon environment on Treg under homeostatic conditions as well as after alloBMT.

To better understand organ-specific and therapy-related gene expression patterns and to assign functional categories to differentially expressed genes, we next applied graph-based correlation clustering to the entire set of Treg expression profiles. This approach allowed the identification of gene sets (co-expression clusters) that share similar expression across all Treg samples. The main component of the gene-to-gene network (including 3829 genes) is shown in Fig. 3d, with the coloring of nodes (genes) indicating individual co-expression clusters and with top enriched functional annotations assigned to each cluster (detailed results of the functional annotation are provided in Supplementary Fig. 3m). The comparison of network and gene

expression data (the corresponding heatmap is shown in Fig. 3e) highlights the pronounced effect of organ-specific environments across different treatment groups as well as functional differences across Tregs in different environments. For example, the colon specific co-expression cluster 1 was enriched for overlapping gene sets involved in TNF and IL-2 signaling, with the latter known to be active in colon-resident cells already under steady-state conditions[32], while BMT-induced cluster 3 contained genes involved in T cell activation and alloreactivity. Other co-expression clusters with prominent functional associations included cluster 4 (comprising cell cycle-associated genes) or cluster 9 (enriched for IFN response genes), which were prominent in Treg re-isolated from liver and spleen of alloBMT recipients. In line with the functional annotation, we also observed the enrichment of specific transcription factors within co-expression clusters (Fig. 3f), suggesting their involvement in the environment-specific rewiring of Treg transcriptional programs. For example, cluster 1, corresponding to strictly colon-enriched genes, included transcription factor-encoding genes such as *Ets2 and Runx3*, which were previously implicated in colon-resident Treg-specific transcriptional networks[33], and *Stat3, Stat5a/b, Klf6* or *Bcl6*, which are associated with IL-2 signaling, while the more BMT-specific cluster 3 included *Batf, Gata3, Nfil3, Tbx21* or *Hif1a*, which are all associated with T cell activation.

In summary, the co-expression network, integrating BMT as well as noBMT samples, clearly indicates that both allo- and poly-expanded donor Treg rapidly adapt to organ-specific environments. It also highlights organ-specific functional differences of Treg related to BMT and the presence or absence of Tconv.

## GvHD induces protective gene expression programs in organ-infiltrating donor Treg

To further delineate the effect of GvHD from that of confounding factors (i.e., irradiation) on phenotype and function of donor Treg upon transfer into alloBMT recipients, we next focused on differences between Treg in GvHD and noGvHD samples. Scatter plots in Fig. 4a–c highlight the impact of co-transplanted Tconv (GvHD) on Treg expression signatures across the individual organs. GSEA (shown in Supplementary Fig. 4a–c) revealed the significant GvHD-dependent enrichment of type-I interferon response genes in spleen- and liver-derived Treg (upregulated signature genes are labeled in green in Fig. 4a, b). In line with the profiles observed in co-expression Cluster 4, genes associated with proliferation were specifically induced in GvHD spleens (labeled in purple in Fig. 4a). Most strikingly, however, donor Treg co-transplanted with Tconv, thus under GvHD conditions,

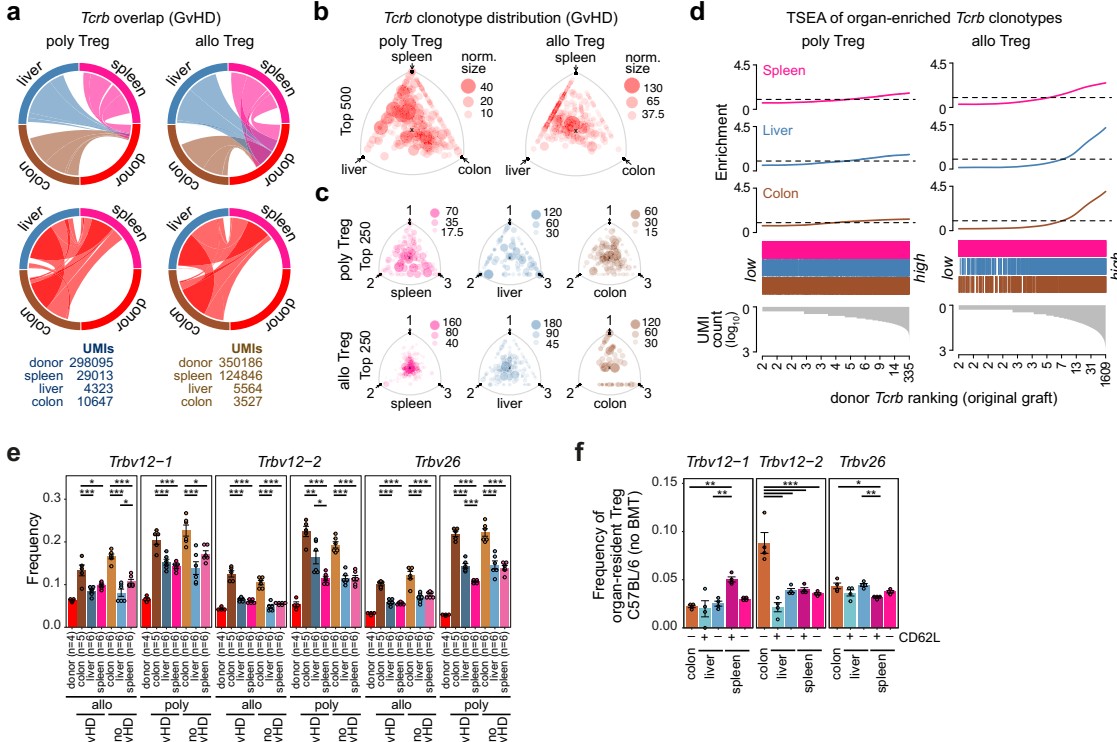

**Fig. 5 | Major overlap of donor Treg TCR repertoires in recipient organs after BMT. a** Circos plots presenting the donor Treg TCR repertoire overlap in the prophylaxis set-up (GvHD, experiment 1) between graft (donor, poly or allo) and recipient organs (top panels) or between organs (bottom panels) across the three recipients of one donor Treg product. Bands represent the fractions of UMI overlaps of pooled organs. Total numbers of *Tcrb* UMIs (representing single mRNA molecules) are given below the plots. **b** Corresponding barycentric distributions (as explained in Supplementary Fig 5m) of *Tcrb* clonotypes between organs. Data of recipient organs were merged. **c** Barycentric distributions of *Tcrb* clonotypes between replicates of individual organs. **d** TCR set enrichment analysis (TSEA) of donor Treg *Tcrb* clonotypes detected in host organs after BMT (prophylaxis) across frequency-ranked clonotypes of the originally grafted donor polyTreg or alloTreg products. In (**a**–**d**) data are from one of two independent experiments with independent donor Treg products (allo and poly, respectively) simultaneously

transplanted into three recipient mice each. Corresponding results for the second experiment and the no GvHD set-up are shown in Supplementary Fig. 5. **e** Frequencies of three colon-enriched *Trbv* segments across in vitro-expanded allo and polyTreg (donor) and donor Treg re-isolated from the indicated organs. Bars represent mean values ± SEM of the indicated number of independent recipient animals. Individual data points are shown as dots. **f** Frequencies of the same *Trbv* segments as in (**e**) across Treg isolated from the indicated organs of non-transplanted C57BL/6 mice. Bars represent mean values ± SEM of *n* = 4 independent isolations pooling cells from 2–3 animals each. Individual data points are shown as dots (**e**, **f**). Significant differences between *Tcrb* distributions in donor Treg re-isolated from the indicated organs are indicated on the top (one-way ANOVA with Tukey's post hoc test). **e**, **f** *$p < 0.05$, **$p < 0.01$, ***$p < 0.001$. **a**–**f** Source data and exact *p* values (**e**, **f**) are provided as a Source Data file.

showed a significantly enhanced expression of genes associated with suppressive function (labeled in black in Fig. 4a–c, and schematically depicted in Fig. 4d). This was detected in both Treg populations (allo and poly) and across all organs (again most pronounced in colon) and thus confirms their functional activity in GvHD prophylaxis. A subset of these suppressive signature genes was already present in resident colon Treg (noBMT), including *Ctla4*, *Tgfb1*, *Nt5e* (coding for ecto 5'-nucleotidase or CD73), *Fgl2* (fibrinogen-like protein 2) and *Il10*. This is consistent with their mixed composition (tTreg and pTreg) and their homeostatic function at such barrier sites under steady-state conditions, as outlined above. In addition, GvHD induced the NK/CD8+ T cell-associated effector molecules granzyme A and B and perforin (*Gzma*, *Gzmb* and *Prf1*) in donor Treg (in all organs, but again most evident in colon) suggesting that one of their protective mechanisms in GvHD involves the killing of donor Tconv or APC[34–36] (for expression profiles of individual genes across all samples see Supplementary Fig. 4d).

### The TCR repertoire of transplanted donor Treg comprises dominant organ-spanning as well as smaller organ-restricted clonotypes

Our transcriptome analyses highlighted the plasticity and functionality of in vitro expanded donor Treg, which rapidly adopted organ-specific

gene-expression profiles and initiated "protective" programs in the presence of GvHD-inducing Tconv. To evaluate the impact of TCR-specificities on their seeding and function, we analyzed the distribution of clonotypes and their families (Vβ genes) in donor Treg after alloBMT (applying high-throughput sequencing of TCRα/β-chain CDR3 region-specific RACE-PCR fragments). We first compared relative clonotype frequencies of re-isolated donor Treg to those of the original Treg graft or between organs of the three recipient mice that received the same Treg product (Fig. 5a–c). As shown in Fig. 5a for *Tcrb* in Treg co-transferred with Tconv (GvHD), we found that the proportion of transplanted Treg clonotypes seeding spleen, liver or colon was smaller for poly compared to alloTreg (Fig. 5a, upper panels; results of additional GvHD and noGvHD experiments are shown in Supplementary Fig. 5a–l), reflecting the differences in TCR repertoire diversity and specificity between the two Treg products. The clonal overlap of donor-derived Treg between organs, however, was comparable between allo and polyTreg (Fig. 5a, lower panels). The distributions of the top abundant clonotypes were visualized using barycentric triangle plots, where the position of a bubble in the triangle reflects the respective organ contribution to the clonotype and bubble size corresponds to the normalized size of each clone (see also Supplementary Fig. 5m for further explanation). Major clonotypes were frequently identified in all three organs of individual recipients (Fig. 5b) as well as

across respective organs of all three separate recipients (Fig. 5c). Again, this was more evident for alloTreg, suggesting TCR-driven rather than stochastic distribution of donor Treg clonotypes. In addition, re-isolated alloTreg were enriched for clones already expanded during culture (Fig. 5d), which was less pronounced for polyTreg. The observed clonotype distribution patterns indicated that expansion and infiltration of donor-derived Treg into recipient organs is indeed mainly driven by alloreactivity.

Repertoire diversities of re-isolated donor allo and polyTreg were comparable in all organs and for both BMT settings (GvHD and noGvHD), but in all cases lower than in the original grafts (Supplementary Fig. 5n), demonstrating the in vivo selection of clonotypes. To further investigate potential organ-specific shifts in TCR repertoires, we next compared TCR repertoires of in vitro expanded donor Treg and cells re-isolated from recipient organs on the level of Vβ (*Trbv*) gene segment usage. In the colon, we noted a significant enrichment of three particular *Trbv* segments (*Trbv12-1*, *Trbv12-2*, *Trbv26*) which was more pronounced in poly than in alloTreg (Fig. 5e). Clones with these segments were generally under-represented in the in vitro expanded Treg products (Supplementary Fig. 5o, top panels) yet almost evenly distributed across TCR-frequencies of colon-infiltrating donor Treg (Supplementary Fig. 5o, bottom panels). Their low frequency upon ex vivo expansion but organ-specific selection after transfer was clearly distinct from the dominant donor Treg clones identified across all organs and recipients. This suggests that they specifically respond to local tissue, microbial or dietary antigens that are presented either per se (*Trbv12-2* is also enriched in resident colon Treg, Fig. 5f) or only in response to conditioning-induced organ damage (*Trbv12-1*, *Trbv26*, which are only enriched in alloBMT colon).

Collectively, the observed distribution of clonotypes and Vβ gene segments suggests that the seeding of donor Treg had a dominant alloreactive component shared across organs. An additional tissue-specific component was detected in colon. Its contribution to the repertoire was higher in polyTreg as compared to alloTreg, suggesting that the latter lose respective clonotypes during in vitro expansion.

**Transcriptional rewiring during organ-infiltration is largely independent of the TCR**

While our bulk transcriptome and repertoire analyses already revealed the tissue imprint as well as the clonotypes of re-isolated Treg from different organs, the respective technologies are unable to match α- and β-chains or to associate clones with their respective transcription profiles. To determine whether the transcriptional adaption in separate organs occurs on the level of individual Treg clones, and to uncover potential differences between ubiquitous and tissue-specific clones, we performed single-cell RNA sequencing (scRNA-seq) combined with TCR sequencing of donor Treg from the GvHD prophylaxis model using the 10X Genomics platform. As observed in our bulk analysis, transcriptomes of colon-derived donor Treg showed typical organ features that were distinct from liver and spleen, which occupied overlapping clusters (Fig. 6a). Replicates evenly contributed to individual clusters (Supplementary Fig. 6a) and their defining gene signatures and markers are shown in Fig. 6b, c (and in more detail in Supplementary Fig. 6b). As shown in Fig. 6d, several of these single-cell clusters overlapped with gene signatures of co-expression clusters derived from bulk data, such as single-cell clusters 8 and 11 which feature a prominent IFN response signature that is also seen in co-expression cluster 9 in Fig. 3d. Organ specific gene signatures were generally comparable between bulk and single-cell experiments, as exemplified by genes encoding transcription factors, including *Ets2*, *Klf6*, or *Rora*, which were highly expressed in Treg re-isolated from colon (single-cell clusters 1&3, co-expression clusters 1&2), but not in liver or spleen.

To match gene expression with clonotype data, TCR repertoires were analyzed on single clone level (defined by the presence of a single α- and β-chain). TCR diversity declined (as expected) in re-isolated Treg compared to the input donor population, with colon still retaining the highest diversity amongst the analyzed tissues (Supplementary Fig. 6c). A substantial fraction of TCR clones overlapped between organs of recipient mice as well as with the original graft (Fig. 6e), with spleen and liver repertoires showing the highest similarity (Supplementary Fig. 6d). As observed in bulk *Tcrb* analyses, major TCR clones were frequently identified in all three organs (Fig. 6f, left panel), and across organs of individual recipients (Fig. 6f, other panels). Figure 6g shows an example of such a TCR clone (*Trav16D-DV11/Trbv3*) that is present in all organs (as evidenced by the colored dots in the UMAP frame) of all recipients. The position in the UMAP suggests that each Treg cell expressing this particular TCR adopted an organ-specific gene expression signature depending on their location (as also exemplified by typical colon signature genes in Fig. 6h) despite sharing the same TCR. This proves again that tissue-specific transcriptional rewiring occurs in vivo that is largely independent of TCR-mediated signals.

In addition to the more ubiquitously appearing clones (examples given in Fig. 6i, top panels), we also observed smaller-sized, organ-enriched clones (examples given in Fig. 6i, bottom panels), among them clones in the colon carrying one of the three *Trbv* segments identified in our bulk analyses (Supplementary Fig. 6e, f). This prompted us to specifically dissect sub-clusters of Treg re-isolated from this organ. Focusing on colon cells only, we identified eight sub-clusters (shown in Fig. 7a) defined by specific marker genes (as shown in Fig. 7b and in more detail in Supplementary Fig. 7a). Those were enriched for particular pathways (as shown in the top panels of Fig. 7c), including glycolysis (sub-cluster 2) or NF-κB signaling (sub-cluster 3). Expression profiles of additional cluster-defining or Treg marker genes are shown in the bottom panels of Fig. 7c and Supplementary Fig. 7b. Apart from genes involved in glycolysis (such as *Pgk1*, *Pgam1*), Treg in sub-cluster 2 expressed high levels of *Dgat1*, encoding one of the rate-limiting enzymes in the production of triglycerides and thus confirming the high metabolic activity of cells within this cluster. Interestingly, this sub-cluster was also enriched for the NLT-Treg marker *Klrg1*. NF-κB-activated Treg in sub-cluster 3 expressed marker genes such as *Jun*, *Klf6*, *Zfp36* and *Irf4*, while Treg expressing effector molecules such as *Prf1*, *Fasl*, *Entpd1* (coding for CD39) and *Gzma* were predominantly found in central sub-clusters 1 and 4. A separate Treg population expressing *Klrg1* together with *Areg*, *Il1rl1* (ST2) and *Gata3* and thus resembling a previously described NLT-Treg subpopulation with tissue-repair function[37] resided in sub-cluster 5. In summary, the clustering of single-cell expression profiles indicates that the donor Treg cells populating the colon in alloBMT recipients are metabolically and functionally diverse.

To reveal potential relationships and developmental trajectories across donor Treg re-isolated from host colon, we performed RNA velocity analysis. The projection of RNA velocities onto the UMAP embedding predicted a predominantly unidirectional transition from repair-type donor Treg in sub-cluster 5 to NF-κB activated Treg in sub-cluster 3 (Fig. 7d). This is mirrored by expression changes in key genes along the trajectory, such as *Tnfrsf9* (encoding 4-1BB; see Fig. 7c).

Finally, we studied the distribution of the three colon-enriched *Trbv* segments identified in our bulk analyses (*Trbv12-1*, *Trbv12-2*, *Trbv26*) among the eight colon sub-clusters. As shown in Fig. 7e, cells expressing one of the three Trbv segments were particularly enriched in the metabolically active sub-cluster 2, but virtually absent from the "activated" clusters 1, 3 and 4 and showed a significantly different gene expression profile than cells expressing any other *Trbv* segment, with *Klrg1* being one of the top markers (Fig. 7f).

## Discussion

Despite an overall improvement in incidence and mortality rates during the last 30 years, acute GvHD remains a major challenge of allo-

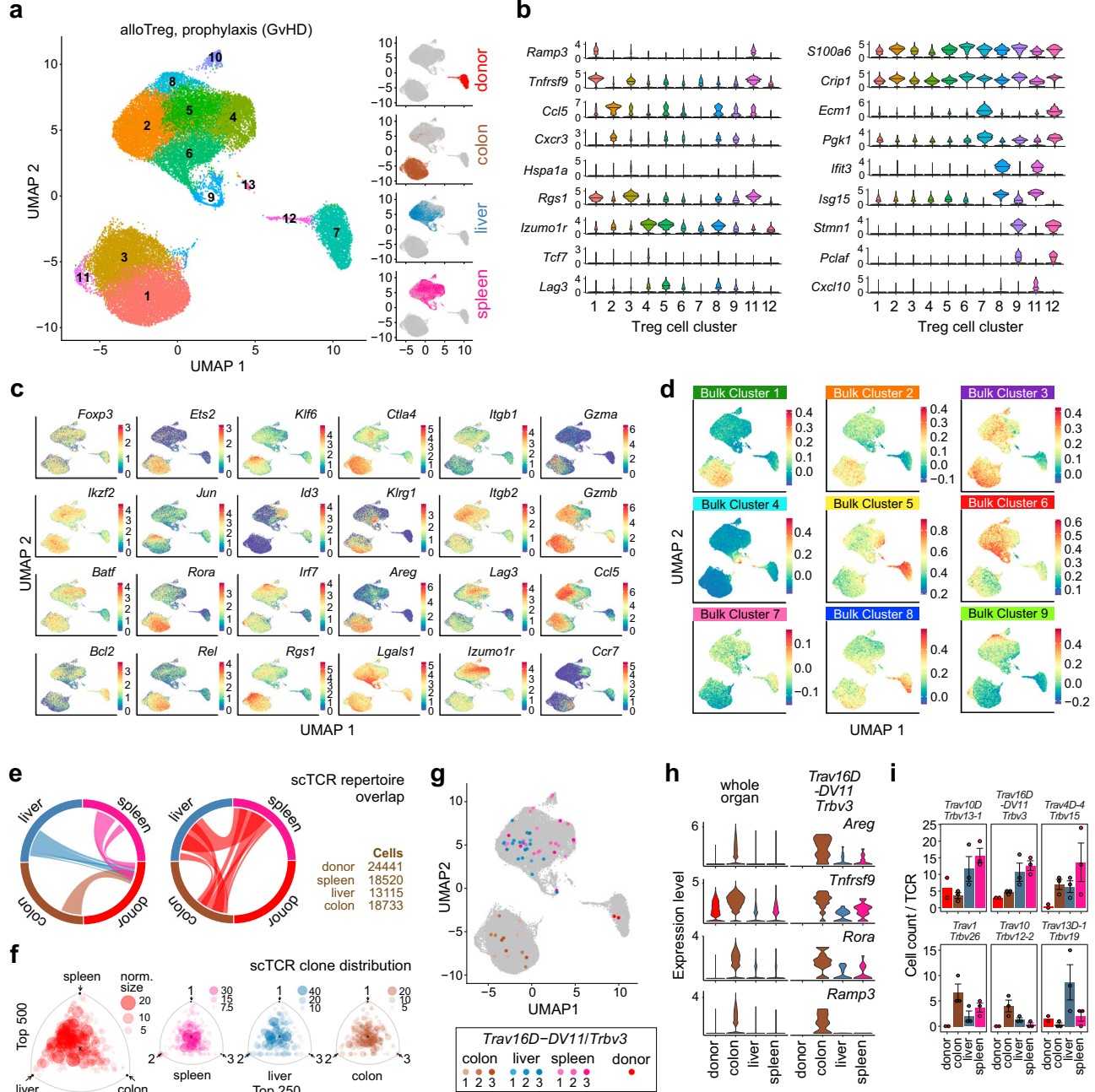

**Fig. 6 | TCR-independent transcriptional re-wiring of donor Treg in recipient organs after BMT. a** UMAP embedding of 37,457 donor Treg cells (including the original Treg product from allogeneic expansion (donor), as well as donor Tregs isolated from recipient organs from $n = 3$ animals receiving the same graft) analyzed by scRNA-seq and colored by unsupervised clustering (Louvain). Four small representations of the same embedding show cells colored by their origin. Clusters 10 and 13 contained contaminating myeloid (cluster 10) or erythroid signatures (cluster 13) and were excluded from further analyses. **b** Violin plots showing the expression distributions of the top cluster-defining genes across Treg clusters. **c**, **d** Expression of selected marker genes (in **c**) or gene signatures derived from co-expression network analysis of bulk RNA-seq data as shown in Fig. 3d (in **d**) across the UMAP embedding. Expression levels are in log2 scale and color coded as shown in the corresponding legends. **e** Circos plots presenting the TCR repertoire overlap

between graft (donor) and recipient organs (left panel) or between recipient organs (right panel). Bands represent the fraction of overlap. Total numbers of cells with informative TCRs are given on the right. **f** Barycentric distributions of TCR clones between organs (left panel) or between replicates of individual organs (three right panels), as described in Fig. 5. Only the top 500 (left panel) or top 250 clones (three right panels) are shown for clarity. **g** Cells expressing one particular TCR (as indicated) are marked in the UMAP embedding (as shown in (**a**)) and color-coded by their origin. **h** Violin plots representing the expression of four colon Treg signature genes across whole organs (left panel) or across cells expressing the indicated TCR (right panel). **i** Cell counts for individual TCRs representing more ubiquitous (top) or organ-enriched distributions (bottom). Bars represent the mean of $n = 2$ donors or mean values ± SEM of $n = 3$ recipients. Individual data points are shown as dots. **e**, **f**, **i** Source data are provided as a Source Data file.

HSCT[38]. In experimental studies, the co-transfer of donor-derived CD4+CD25+Foxp3+ Tregs with the stem cell graft emerged as a promising cell-based strategy for aGvHD prophylaxis[12,14,16,19,39]. Recently, first clinical trials with engineered grafts from haploidentical as well as

matched related and unrelated donors have been conducted and confirmed the validity of the concept[13,40–42].

Expansion of Treg either in vitro prior to or in vivo after application has been explored by several groups in the fields of allogeneic

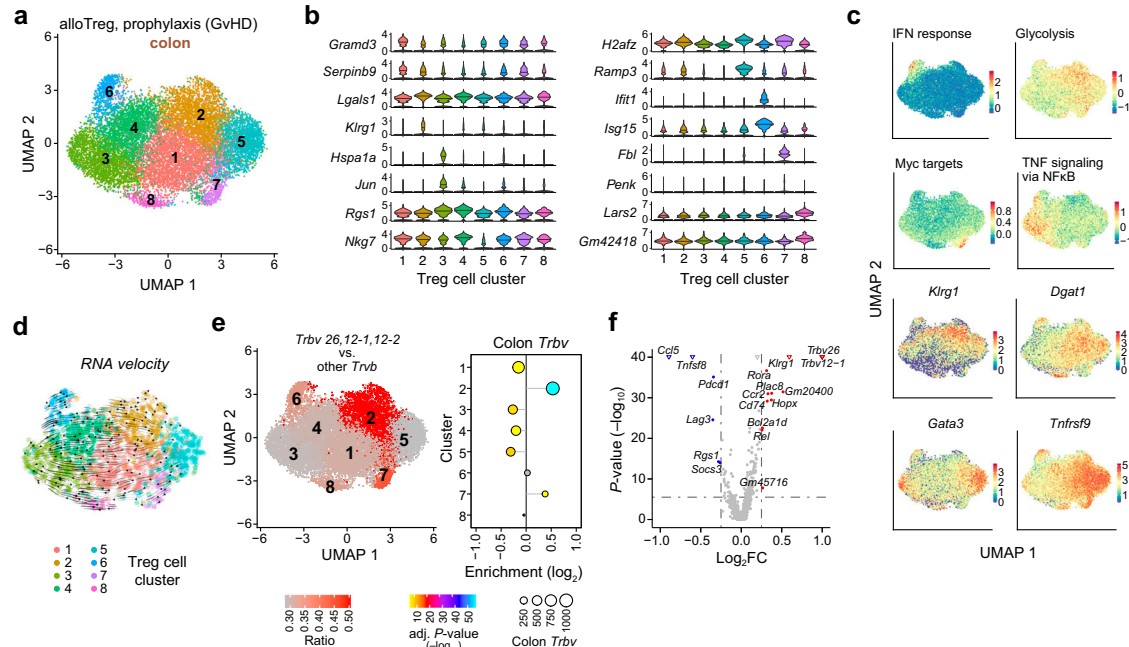

**Fig. 7 | Colon-specific reprogramming of donor Treg after BMT. a** UMAP embedding of only colon-derived donor Treg cells (from *n* = 3 animals receiving the same graft) analyzed by scRNA-seq and colored by unsupervised clustering (Louvain). **b** Violin plots showing the expression distributions of marker genes across colon Treg clusters. **c** Expression of gene signatures (upper panels; genes listed in Supplementary Table 1) or selected marker genes (lower panels) across the UMAP embedding. Expression levels are in log$_2$ scale and color coded as shown in the corresponding legends. **d** RNA velocity estimation is visualized on the pre-defined UMAP embedding. (**e**) Ratios of colon-enriched *Trbv* segments *12-1, 12-2, 26* compared to all other *Trbv* segments across sub-clusters in the UMAP embedding (left

panel) or their enrichment across sub-clusters are shown (right panel; two-sided Fisher's exact test with Benjamini–Hochberg correction). **f** Volcano plot highlighting genes differentially expressed between Treg expressing one of the three enriched *Trbv* segments (*Trbv12-1, 12-2, 26*) or any other *Trbv* segment (Wilcoxon Rank Sum test (two-sided), Bonferroni correction). Hash-dotted horizontal lines mark an FDR value of 0.05 (*y*-axis) and the log2FC thresholds of ±0.25 (*x*-axis). Selected genes are marked by coloring (red:up-/blue:down-regulated), triangles: extremely low *p* values were shifted to the indicated value. **e**, **f** Source data are provided as a Source Data file.

stem cell- and organ transplantation. Most of the protocols applied polyclonal in vitro expansion using CD3/CD28-bound beads or in vivo activation by administration of low-dose IL-2[43–45]. Yet, some investigators suggested superior suppression by allo-antigen-primed as compared to polyclonal donor Treg[16,24,25]. To further explore Treg function in GvHD prevention, we now compared polyclonal Treg cell products to Tregs primed during in vitro expansion by host-APCs (alloTreg). We profiled the transcriptomes and TCR repertoires of the respective Treg populations after in vitro expansion as well as 7 days after adoptive transfer into BMT recipients. We thereby monitored their migratory behavior, their transcriptional profile and TCR-selection and analyzed their influence on co-transferred Tconv cells in MHC-mismatched recipients (as summarized in Fig. 8).

After in vitro culture, both Treg populations showed a stable and comparable expression of key Treg signature genes, such as *Foxp3*, *Ikzf2* (Helios), *Ctla4* or *Tnfrsf9* (4-1BB). This confirms our previous results on human Treg[46] as well as results by Hippen et al.[47], demonstrating a stable phenotype and—unlike Tconv—hardly any signs of exhaustion even after multiple rounds of in vitro stimulation. Furthermore, both Treg products fully retained their proliferative capacity and migratory potential as well as their phenotypic and functional plasticity. Thus, we observed a similar distribution of Tregs in allogeneic recipients as reported before for ex vivo enriched donor Treg[17,19], without major differences between the two in vitro expansion protocols regarding homing patterns or abundance in organs at d7 after transfer. Most interestingly, both Treg populations rapidly underwent tissue-specific transcriptional rewiring upon entering lymphoid and non-lymphoid recipient organs and even adopted the gene expression profile of organ-resident Treg of non-transplanted mice. We and others previously showed that spleen and lymph nodes

contain precursors of tissue Treg under steady-state conditions, that undergo a stepwise differentiation upon their migration to non-lymphoid organs, with final phenotypic and functional adaptation taking place within the respective tissues[9,30,48]. With regard to the clinical use of donor Treg for GvHD prevention or therapy, it is important to note that in vitro expanded Treg obviously retain this adaptation capacity and are thus able to respond to local physiologic or inflammatory cues and to function in a tissue-specific manner.

Despite equal suppressive activity after in vitro culture and comparable homing and adaptation patterns after transfer, alloTreg were more efficient than polyTreg in the early control of aGvHD, which translated to a quicker recovery of allo as compared to polyTreg-treated recipients and a significantly better protection from lethal aGvHD. Although these results are in line with earlier reports comparing polyclonal and recipient-type-specific Treg products[16,25,49,50], we did not observe any inferiority of polyTreg with regard to early activation and expansion or long-term persistence after transfer that could explain these differences. When we analyzed the TCR repertoires of the two differentially cultured Treg populations, we found that poly-Treg retained their TCR repertoire diversity after in vitro expansion (as shown by us before for human Treg[21]), whereas the TCR repertoire of alloTreg was significantly narrowed down, as expected after allo-antigen-driven stimulation. After adoptive transfer into MHC-disparate hosts and re-isolation of donor Treg 7 days after BMT, alloTreg showed a much broader overlap in TCR clonotypes of retrieved vs. infused cells in comparison to poly Treg. Furthermore, dominant TCR clonotypes of alloTreg were more evenly distributed across different recipient organs and also across different BMT recipients receiving the same Treg graft in comparison to polyTreg.

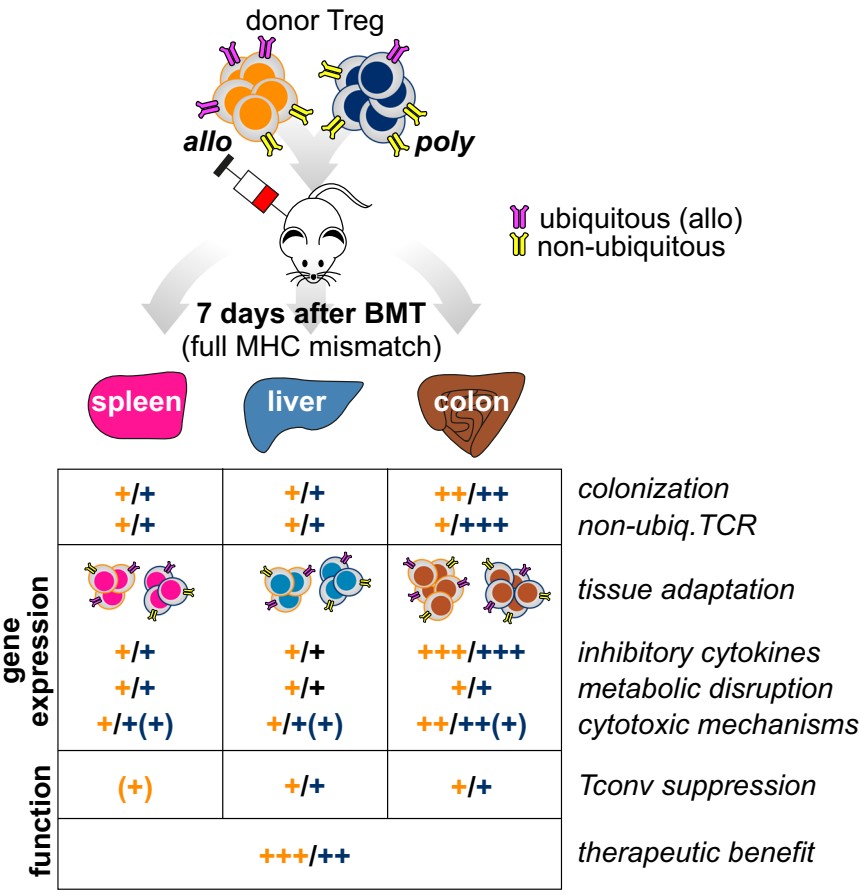

**Fig. 8 | Graphical summary.** Both alloantigen-specific and polyclonal donor Treg expansion protocols generate functional cells that acquire organ-specific gene expression profiles in vivo upon transfer. In addition, both types of expanded Tregs induce genes involved in the suppression of conventional donor T cells (inhibitory cytokines, genes involved in metabolic disruption and cytotoxic genes) and ameliorate GvHD symptoms. The major difference between alloantigen-specific and polyclonal donor Treg is their different TCR repertoire ex vivo and in vivo after transplant, which may contribute to the initial therapeutic benefit and survival advantage of alloantigen-specific Treg.

In contrast, although similar in total numbers, re-isolated poly Treg showed a higher proportion of organ-restricted clonotypes than alloTreg. Noteworthy, these were transcriptionally distinct and less "activated" than ubiquitously distributed clones and appeared primarily in the colon. Their concomitant depletion in allo-antigen-driven expansion cultures suggests that a large fraction of them may respond to antigens specific to the GI-tract, including microbiota and their metabolites, or dietary antigens. The contribution of such GI-tract-restricted clonotypes to aGvHD protection is so far unclear. Whereas our data suggests that they are less engaged in disease control, at least in the initial phase, further studies need to clarify their role in less allo-antigen-driven transplantation models or at later timepoints after alloBMT. Overall, these data suggest that early donor Treg migration and selection in this MHC-disparate model are primarily driven by ubiquitously expressed allo-antigens, which explains the advantage of allo as compared to polyTreg regarding protection efficacy.

Treg employ a range of different suppressive mechanisms that depend on the respective target cell as well as the micromilieu[51] and differ between non-inflammatory and inflammatory conditions[36]. In a recent study by Lohmeyer et al.[52] applying a similar mouse model of GvHD (C57BL/6 → BALB/c), the investigators analyzed donor Treg-mediated Tconv modulation by performing bulk RNA and TCR repertoire sequencing of pooled Tconv re-isolated from recipient spleens and lymph nodes. While confirming their own and our earlier data showing that Treg suppress initial Tconv expansion in these organs[16,24], they now provided evidence that Treg neither diminish the TCR repertoire diversity of the remaining Tconv cells, nor completely block their activation. Rather, Treg seem to use several suppressive molecules, including IL-10, to selectively up- and down-regulate anti- and pro-inflammatory genes. Our study now complements and significantly extends this work with a detailed analysis of donor Treg transferred into alloBMT recipients. We provide the first data set on donor Treg retrieved from non-lymphoid GvHD target organs colon and liver and compared those to Treg transplanted in the absence of GvHD-inducing Tconv as well as to organ-resident Treg in steady-state. Our findings illustrate the tissue-specific adaptation and organ-specific suppression mechanisms of donor Treg in GvHD.

Treg from both in vitro expansion protocols engaged a similar suppressive program after transfer into MHC-mismatched hosts, which included upregulation of various immune homeostasis-associated genes such as *Il2ra* and *Il2rb* (CD25 and CD122), *Entpd1* and *Nt5e* (CD39 and CD73), *Itgal* (Lfa-1), *Icam1*, *Lag3* or *Ctla4*. Strikingly, the presence of aGvHD-inducing Tconv cells led to the additional upregulation of several cytotoxic molecules in the transferred donor Treg, such as granzymes (*Gzma, Gzmb*), perforin (*Prf1*) or Fas ligand (*Fasl*). It has been shown previously that granzyme B-deficient Treg are unable to establish long-term tolerance to allogeneic grafts[53]. Likewise, it was shown that tumor-associated Treg kill DC in tumor-draining lymph nodes in a perforin-dependent manner[34] and to eliminate NK and CD8+ T cells by utilizing granzymes, perforin or the Fas/FasL pathway[54,55]. To what extent such cytotoxic mechanisms are operative in Treg-mediated protection from aGvHD is still under debate[56,57] and requires further investigation. However, their elevated expression clearly distinguished aGvHD samples from resident (noBMT) or

control BMT (noGvHD) samples, suggesting that cytotoxic programs in donor Treg may indeed contribute to control allo-reactive Tconv cells under the highly inflammatory conditions of aGvHD.

The strongest suppression signature was observed in Treg re-isolated from recipient colon. Already under steady-state conditions, intestinal Treg represent a heterogeneous and functionally diverse compartment that contains cells with a 'repair-type' gene expression program, that includes genes encoding Gata-3, IL-10, Areg and ST-2[58,59]. This subpopulation maintains the intestinal stem cell niche[60] and contributes to wound-healing processes at inflammatory sites[37,59]. We recently demonstrated the tissue repair capacity of in vitro expanded polyTreg by analyzing their impact on Paneth cell regeneration in the small intestine of GvHD mice[23]. The now presented single-cell transcriptome analyses reveal that donor Treg re-isolated from host colon display a similar transcriptional profile as their tissue-resident counterparts as early as 7 days after transfer. Thus, transferred donor Treg administered for the prevention of GvHD seem to employ physiological mechanisms of suppression, that seem to vary depending on the spatial, temporal and inflammatory context.

In addition to the known effector molecules mentioned above, we also noted the induction of natural killer (NK) cell receptor genes *Klrc1* (NKG2A) and *Klrd1* (CD94) on donor Treg re-isolated from host colon, particularly in the context of aGvHD. These inhibitory receptors are likely part of a common transcriptional program shared with CD8+ T, NK, and NKT cells. They have also been identified on intestinal γδT cells[61], where their ligation induces release of TGFβ. It is intriguing to speculate that Treg under highly inflammatory conditions show a similar response, thereby inducing conversion of pro-inflammatory Tconv into pTreg and subsequently tipping the balance further toward immune homeostasis.

In conclusion, our data provide a comparative and comprehensive phenotypic and functional analysis of mouse Treg directly ex vivo, after polyclonal and allo-antigen-driven in vitro expansion as well as after transfer into allogeneic recipients with or without concomitant aGvHD. This detailed atlas of gene expression programs and clonotypic selection processes uncovers basic principles of Treg biology and helps to guide future approaches to further improve donor Treg-based interventions in aGvHD.

## Methods

### Mice

Female BALB/cAnNCrl (H-2$^d$) and C57BL/6NCrl (H-2$^b$) wild type (WT) mice were purchased from Charles River Laboratories (Sulzbach, Germany), congenic B6.SJL-Ptprca Pepcb/BoyJ (CD45.1; H-2$^b$; here named "B6-CD45.1") and transgenic FoxP3$^{EGFP}$ mice (on a C57BL/6 genetic background; CD45.2; H-2$^b$; here named "Foxp3$^{gfp}$", kindly provided by B. Malissen[62]) were bred in-house. Donors were 8–12 weeks, recipients 11–16 weeks old at the time of BMT. Mice were held under specific pathogen-free conditions at 22 ± 2 °C, 50 ± 10% humidity on a 12 h day/night cycle and received autoclaved chow and water ad libitum. All animal studies were approved by the Committee on Ethics of Animal Experiments at the Bavarian Government (Ref-No: 55.2-2532-2-430).

### GvHD model

Single-cell suspensions from BM (femora and tibiae of both hind legs) and spleen were prepared. T cells were depleted from BM (T cell depleted bone marrow; TCDBM) using anti-CD90.2 MicroBeads (Miltenyi Biotec). CD4$^+$CD25$^-$ conventional T cells (Tconv) were isolated from splenocytes by depletion of CD25$^+$ cells using anti-CD25-PE (antibodies are listed in Supplementary Table 2) and anti-PE UltraPure MicroBeads (Miltenyi Biotec), followed by enrichment of CD4$^+$ cells from the CD25$^-$ fraction through labeling with anti-CD4 MicroBeads (Miltenyi Biotec). For short-term (d7) FACS and RNA-seq experiments BALB/c recipients were irradiated (8 Gy) on the day of BMT and

transplanted i.v. with 2.5 × 10$^6$ TCDBM and 1 × 10$^6$ Tconv from B6-CD45.1 mice with or without in vitro expanded allo or polyTreg from Foxp3$^{gfp}$ mice at a 1:1 ratio. Recipients of the "no GvHD" group received TCDBM and Treg only. In long-term survival experiments irradiated BALB/c recipients were transplanted with 2.5 × 10$^6$ TCDBM on d0 and 0.25 × 10$^6$ Tconv on d2 for GvHD induction. Mice of the GvHD prophylaxis groups (allo or poly) received additional 0.25 × 10$^6$ in vitro expanded Treg from B6 WT mice on d0. Recipients were monitored daily, body weight and GvHD symptoms assessed twice weekly by non-blinded investigators applying a standardized scoring system[63].

### Cell isolation for cytometric analysis and cell sorting

Single-cell suspensions from BM, spleen and mesenteric lymph nodes were prepared. Erythrocytes were lysed in splenic and BM suspensions. For leukocyte isolation from the liver, the organ was flushed with PBS through the portal vein and the gallbladder removed before excision, transferred to a petri dish and dissected into small pieces. Fragments were suspended in RPMI (Lonza/Biozym)/5% FCS, strained and washed in RPMI w/o FCS before Percoll centrifugation (40%/80%; 800 × $g$/ 20 min/20 °C, w/o break). Leukocytes were resuspended in RPMI/5% FCS, erythrocytes lysed, cells washed in RPMI/10% FCS and kept on ice until further use. For intestinal leukocyte isolation, large intestine was excised, cut into 0.5 cm pieces, incubated twice for 20 min at 37 °C in HBSS/5 mM EDTA/1 mM DTT (Sigma-Aldrich) followed by vigorous shaking to isolate intraepithelial leukocytes (IEL). For lamina propria leukocytes (LPL), the fragments were transferred to HBSS w/o phenol red containing calcium and magnesium with 5% FCS together with enzymes of the Lamina propria dissociation kit (Miltenyi Biotec) and incubated for 30 min at 37 °C. The fragments were further dissociated using the GentleMACS® system (protocol: m_intestine_01; Miltenyi Biotec), strained, washed, and pooled with IEL before Percoll (Sigma-Aldrich) centrifugation (30%; 512 × $g$/15 min/20 °C). For baseline (no BMT) colon Treg sorts IEL were not included (no Tregs in untreated animals) and colonic LPL were further enriched using anti-CD45 MicroBeads and the AutoMACS® system (Miltenyi Biotec). Splenocytes were enriched for CD25$^+$ cells by MACS® using anti-CD25-PE and anti-PE UltraPure MicroBeads before sorting. Cells were further stained for CD8, CD4 and CD62L (also CD45 for colon-derived cells) and Treg were sorted on a FACSAria™ IIu (with FACSDiva™ v8.0.1) or a FACSAria™ Fusion (with FACSDiva™ v8.0.3; both Becton Dickinson). From spleen and liver, CD62L$^+$ and CD62L$^-$ subpopulations were isolated, from colon, only CD62L$^-$ cells were obtained (sorting strategy shown in Supplementary Fig. 3a). For re-isolation of donor Treg on d7 after transplantation, single-cell suspensions from spleen, liver and colon were prepared as described, stained for H-2K$^b$, CD45.1, CD45.2, TCRβ and CD4 and sorted on a FACSAria IIu™ or a FACSAria™ Fusion (sorting strategy see Supplementary Fig. 3b).

### Flow cytometry

Staining was performed in PBS/2%FCS with anti-CD16/CD32 antibodies to block FcR-binding and DAPI (Sigma-Aldrich) or Fixable Viability Dye (Thermo Fisher) to exclude dead cells (antibodies are listed in Supplementary Table 2). For intracellular markers, the Foxp3/Transcription Factor Staining Buffer Set® (eBioscience) was used. Multicolor FACS staining for surface markers was performed using Brilliant Stain Buffer (BD Biosciences) where applicable. Data were acquired on a FACSymphony™ A5 SORP (with FACSDiva™ v9.1; Becton Dickinson) and analyzed using FlowJo® (v10.7.1 or v10.8.0; Treestar Inc).

### In vitro Treg expansion

Splenocytes were stained with anti-CD25-PE and anti-PE UltraPure MicroBeads and enriched for CD25$^+$ cells by MACS® (Miltenyi Biotec). The CD25$^+$ fraction was further stained for CD4, CD8 and CD62L and Treg were sorted on a FACSAria™ IIu or FACSAria™ Fusion (Becton Dickinson) as CD4$^+$CD25$^{high}$CD62L$^+$ cells; purity of sorted cells was

>98%. Polyclonal in vitro expansion of Treg was performed as described[23]. In brief, sorted Treg were cultured in DMEM (high glucose, Gibco/Invitrogen) with 10% FCS, 2mM L-glutamine, 10 mM HEPES, 1% NEAA (PAN Biotech), 50U/ml penicillin, 50 μg/ml streptomycin and $5 \times 10^{-5}$ M 2-mercaptoethanol (Gibco/Invitrogen) (cDMEM) in 96-U-well plates ($1 \times 10^4$/well) and stimulated with CD3/CD28-beads (Treg-Expansion Kit, Miltenyi Biotec, 4 beads/cell) and rhIL-2 (2000 U/ml; Proleukin®, Chiron). Cultures received 100 μl/well fresh cDMEM with IL-2 on d4, were restimulated on d7 with 1 bead/cell after transfer into 24-well-plates ($1 \times 10^6$ cells/ml/well), fed and split on d10 as needed and harvested on d12. For allospecific in vitro expansion sorted Treg were seeded at $5 \times 10^4$/well in cDMEM together with $2.5 \times 10^4$/well (d0) or $1 \times 10^4$/well (restimulation on d7) anti-CD11c MicroBead-enriched (Miltenyi Biotec) and irradiated (30 Gy) CD11c+ DC from BALB/c mice. Cells received fresh medium on d4 and d10 and were harvested on d12.

### Treg suppression assay
CD4+CD25− Tconv cells were isolated from splenocytes of syngeneic C57/BL6(CD45.1+) mice as described above, labeled with CFSE (Sigma-Aldrich) and served as responder cells (Tresp). T-cell-depleted C57/BL6(CD45.1+) splenocytes were also obtained by MACS® (CD90.2-microbeads, Miltenyi Biotec) and served as APC after irradiation with 30 Gy. Tresp (50,000 cells/well) were seeded with APC (100,000 cells/well) and titrated numbers of in vitro expanded C57BL/6 WT (CD45.2+) allo- or polyTreg to yield Treg:Tresp ratios of 1:1, 1:4 and 1:16 in round-bottom microtiter plates. Cells were stimulated via anti-CD3 antibodies (NA/LE clone 145-2C11, BD Biosciences at 0.4 μg/ml) for 3 days at 37 °C, 5% $CO_2$, harvested, stained with CD4-BV605 and CD45.2-PB and run on a BD FACS Fortessa. Tresp were identified as CD4+CD45.2−. Tresp proliferation and Treg-mediated suppression were determined by CFSE dilution and analyzed using the FlowJo® "Proliferation" tool.

### Isolation of RNA, library preparation and sequencing
Total RNA was isolated using the RNeasy Mini ($>5 \times 10^5$ cells) or RNeasy Micro ($<5 \times 10^5$ cells) Kits (Qiagen). RNA concentration and quality were measured using RNA ScreenTape assays (Agilent), depending on the expected yield. RNA-seq libraries were prepared using the SMART-Seq® Stranded Kit (Takara) following the manufacturer's instructions. Fragmentation times and second PCR cycle numbers were adjusted depending on sample quality and abundance, respectively. For some samples, first PCR products were pooled to minimize carry-over losses. The concentration of dsDNA libraries was measured with the Qubit™ dsDNA HS Assay Kit (Thermo Fisher Scientific). DNA fragment size distribution was assessed using the High Sensitivity D1000 ScreenTape Assay (Agilent). Sequencing was performed using the Illumina NextSeq550 sequencer and libraries are summarized in Supplementary Tables 4–7.

### TCR repertoire sequencing library preparation
Total cellular RNA was obtained as described in section "RNA-seq library preparation". TCR repertoire sequencing libraries were prepared using a protocol adapted from ref. [64]. For the generation of 5′RACE-ready cDNA total RNA ranging from 10–200 ng (low input) or 200–1000 ng (high input) in a total of 10 μl RNAse-free water was combined with 1 μl of 12 μM or 25 μM 5′-CDS Primer A in a total volume of 11 μl, respectively. Reactions were incubated at 72 °C for 3 min followed by 42 °C for 2 min for poly-dT primer annealing. Next, 1 μl of 10 μM (low input) or 50 μM (high input) TCR_UMI_Smarter template-switching oligo (TSO) was added at room temperature. RT Buffer Mix (5x First-Strand-Buffer Mix, 20 mM dNTP, 100 mM DTT) was freshly prepared on ice before it was supplemented with RNase-OUT™ Recombinant Ribonuclease Inhibitor (40 U/μl) (Thermo Fisher Scientific) and SMARTScribe Reverse Transcriptase (100 U/μl) (Takara) at room temperature. A volume of 8 μl of the resulting reverse transcription master mix was added to each reaction. After

homogenization, the 20 μl reactions were incubated at 42 °C for 90 min followed by 70 °C for 10 min and put on hold at 10 °C in a hot-lid thermal cycler. Following reverse transcription, excess TSO was digested using Uracil-DNA Glycosylase (UDG) (New England Biolabs) by adding 4 μl of UDG master mix adjusted for TSO input amount and successive incubation at 37 °C for 60 min. The master mix for PCR1 was prepared on ice using the Advantage 2 PCR Kit (Takara) along with primers specific for the constant regions of the Trac and Trbc loci as well as a universal forward primer specific for a binding sequence introduced via the TSO. In total, 24 μl of RACE-ready cDNA were combined with 26 μl of PCR1 master mix and subjected to a two-step PCR program (95 °C, 1 min; 95 °C, 30 s and 68 °C, 70 s (x times); 68 °C, 7 min; 8 °C, hold). Depending on the RNA input amount, PCR1 cycle numbers ranged from 16 to 33. The 50 μl PCR product was purified with 27.5 μl of AMPure XP beads (Beckman Coulter) according to the manufacturer's instructions. After elution in 23.5 μl of Monarch® DNA Elution Buffer (New England Biolabs), reactions were split into 10 μl technical replicates for successive extension PCR (PCR2). PCR2 master mix was assembled using the Advantage 2 PCR Kit (Takara) and gene-specific nested primers (Trac, Trbc loci) introducing Illumina Nextera XT i5 indices (S501, S502, S503, S508, S517, S520, S507), distinguishing Tcra from Tcrb libraries within each reaction. After addition of 36 μl PCR2 master mix, 4 μl of Nextera XT Index Kit v2 (Illumina) adapters were used for indexing libraries individually. Libraries were amplified using the same program as in PCR1 with 18–23 PCR2 cycles. Library purification was carried out using AMPure XP beads (Beckman Coulter) at a volume ratio of 0.8 according to the manufacturer's instructions. After final elution in 17 μl of Monarch® DNA Elution Buffer (New England Biolabs), concentration was measured with the Qubit™ dsDNA HS Assay Kit (Thermo Fisher Scientific) and fragment size profile was assessed using the High Sensitivity D1000 or D5000 ScreenTape Assay (Agilent). Sequencing (paired-end 300 bp) was performed on the Illumina MiSeq system. Oligonucleotide sequences are summarized in Supplementary Table 3, Libraries are summarized in Supplementary Tables 4–7.

### scRNA-seq and scTCR-seq library preparation and sequencing
Freshly sorted, re-isolated Treg or donor Treg were loaded on the Chromium Single-Cell Controller (10x Genomics) using the Single-Cell 5′ Library & Gel Bead Kit v2 (10x Genomics #120237). cDNAs were amplified using 12–14 cycles of PCR. ScRNA-seq libraries were constructed using 13–16 cycles of PCR. Products were purified using Ampure XP beads and quality was controlled using Agilent Tapestation. Samples were sequenced (single-reads, S1 flow cell, 100 bp) on the Illumina NovaSeq™. ScTCR-seq libraries were generated from the same cDNAs using the Single-Cell V(D)J Enrichment Kit, Mouse T Cell (10x Genomics) and sequenced on the Illumina NextSeq 550 with 150 cycles single-read sequencing. Libraries (scRNA and scTCR, along with basic QC data) are summarized in Supplementary Tables 7 and 8.

### Analysis of flow cytometry data
Flow cytometry data were analyzed with FlowJo® v10.7.1/v10.8.1 (Treestar Inc., Ashland, OR), including pseudocolor (Supplementary Figs. 1e and 2d) and dot plots in Supplementary Fig. 3a, b. For clustering and embedding of multiparameter flow data (as presented in Supplementary Fig. 2a) FlowJo® plugins DownSample_v3.3 and UMAP_v3.1 were used. The UMAP plot in Supplementary Fig. 2a was created on the basis of 27 samples from spleen, liver and colon (all $n = 3$) from 3 experimental groups: alloTreg prophylaxis, polyTreg prophylaxis and GvHD control and 3 independent experiments, with every sample representing a cell pool of 2-3 mice. Cells were stained for CD45.1, CD45.2, H-2k^b, TER119, TCRβ, CD8α, CD4, CD19, Nkp46, CD11b, Ly6C, Gr-1 and CD25. Dead cells were excluded with DAPI. Samples were gated on live single CD45.1+TER119− and CD45.2+TER119− cells, downsampled to 15,000 cells each and assembled into one file. Plots

representing relative or absolute cell count data presented in Figs. 1b, d and 2b, c and Supplementary Figs. 1c and 2b–d were generated using the ggplot2 package (v3.3.3) in R. Data that passed the normality test (shapiro.test function in R) were analyzed using two-tailed Student's *t* test for two groups (BM and mLN panels in Fig. 2b and Supplementary Fig. 2b, or one-way ANOVA with Tukey's post hoc analysis for more than 2 groups (other panels in Fig. 2b–d and Supplementary Fig. 2b, c). Data that were not normally distributed (liver panel Fig. 2b and spleen and liver panels in Supplementary Fig. 2c) were analyzed using a Kruskal–Wallis test with post hoc analysis using Dunn's test (using the kwAllPairsDunnTest function in the PMCMRplus R package, v1.9.3). The histogram shown in Supplementary Fig. 1b was generated using FlowJo 10.8.1. (Treestar Inc.,Ashland, OR).

### Analysis of survival and GvHD score data

Survival data were analyzed using the survival package (v3.2-13) in R. Pairwise comparisons were performed using the log rank test provided by the pairwise_survdiff function. Survival curves shown in Fig. 2d were plotted using the ggsurvplot function of the survminer package (v0.4.9). Clinical GvHD scores presented in Fig. 2e were analyzed for significant differences between groups (per time point) using one-way ANOVA with Tukey's post hoc analysis and plotted using ggplot2. Graphs were assembled, formatted, and labeled in Adobe Illustrator (v25.2.1).

### RNA-seq analysis

Base-calling and demultiplexing of sequencing reads was carried out using the bcl2fastq Conversion Software (v1.8.4) provided by Illumina. Sequencing reads were mapped to the mouse genome (Gencode Release M16, GRCm38.p5, primary assembly) using STAR v2.5.3a. The genome index incorporated gene annotation from GENCODE Mouse release M16. Tables of raw uniquely mapped read counts per gene were generated during mapping using the built-in --quantMode GeneCounts option in STAR. Differential expression analysis was carried out on raw gene counts using edgeR (v3.34.0) in R (v4.1.0). Pairwise comparisons of indicated data sets were done using the quasilikelihood *F*-test (glmTreat function in edgeR) against a globally applied 1.5-fold change threshold. Scatter plots of edgeR results in Figs. 1e, 3b, 4a–c and Supplementary Fig. 3c, g were generated using the ggplot2 (v3.3.5) and ggrepel (v0.9.1) packages in R and labels were edited in Adobe Illustrator (v25.2.1). The heatmap of signature genes shown in Fig. 1f was generated using the heatmap.2 function of the gplots R package with log2-transformed RPKM values extracted using the rpkmByGroup function in edgeR. Dimensionality reduction based on the Uniform Manifold Approximation and Projection (UMAP) algorithm (as shown in Fig. 3a) was done using the umap package (v0.2.7.0) and visualized using ggplot2 in R. Gene set enrichment analyses of defined gene sets were performed using the function fry of the limma package (v3.48.1) in R. Gene set enrichment analyses using HALLMARK and KEGG gene sets (retrieved from bioinf.wehi.edu.au/MSigDB/v7.1/) were performed using the limma function camera. Barcode representations of gene set enrichment analyses (as shown in Fig. 3c and Supplementary Figs. 1d, 3d, f, h and 4a–c) were plotted using the barcodeplot function in limma. Bar plots of adjusted enrichment *p* values were plotted using standard functions in R (Supplementary Fig. 3d, f). Graph-based co-expression clustering analysis shown in Fig. 3d was performed using Graphia (v.3.0). The heatmap of genes included in the main co-expression network shown in Fig. 3e used log2-transformed, batch-corrected, normalized and scaled CPM (counts per million) and was generated using the pheatmap package (v1.0.12) in R. Significantly enriched Gene Ontology terms across co-expression clusters were identified using Metascape. Barplots of significance levels shown in Supplementary Fig. 3m were generated using ggplot2 in R. Heatmaps of differentially expressed genes shown in Figs. 3f and 4d used log2-transformed, batch-

corrected, normalized, grouped and scaled CPM and were generated using the heatmap.2 function of the gplots R package (v3.1.1). The heatmap of differentially expressed genes shown in Supplementary Fig. 3i used log2-transformed, normalized, and scaled CPM and was generated using heatmap.2. To generate bar plots of expression levels of selected genes (as shown in Supplementary Figs. 3e, k, l and 4d), library size-normalized and batch-corrected expression data were normalized to transcript length and plotted using ggplot2.

### TCR-seq analysis

Base-calling and demultiplexing based on Illumina barcodes was carried out using the bcl2fastq Conversion Software (v1.8.4). UMI-barcoded and paired TCR-sequencing reads were joined using PEAR (v0.9.11). Demultiplexing and adapter trimming was done using MIGEC (v1.2.9) and consensus sequences of molecular identifier groups (MIGs) were mapped using MIXCR (v3.0.13). Mapped TCR-seq data were further analyzed using the immunarch package (v.0.6.6) in R (v4.0.3) to generate clonotype count tables, calculate *Trbv* gene usage and calculate repertoire diversities. Inverse Simpson indices (as shown in Fig. 1c and Supplementary Figs. 1a and 5n) were calculated using the repDiversity function of immunarch with downsampled data and method "inv.simp" and plotted using the ggplot2 package (v3.3.3). *Trbv* gene usage was determined using the geneUsage function of immunarch with parameters "musmus.trbv",.type = "segment",.ambig = "exc". Results were plotted using ggplot2 (Fig. 5e, f). Circos plots presenting clonotype frequency overlaps (as shown in Fig. 5a and Supplementary Fig. 5a–c) were generated using the circlize R package (0.4.12). Barycentric distributions of clonotype counts in triangles (as shown in Fig. 5b, c and Supplementary Fig. 5d–i) were plotted using ggplot2. Plots presented in Fig. 5d and Supplementary Fig. 5j–l, o were generated using the barcodeplot function in limma, which was modified to allow fixed scales for the worms. The schematic in Supplementary Fig. 5m was generated in Adobe Illustrator (v25.2.1).

### scRNA-seq analysis

Sequencing reads were demultiplexed using cellranger mkfastq (4.0.0)[65] and mapped to the mouse genome using cellranger count (4.0.0). Quality control metrics are provided in Supplementary Table 10. The mouse index was supplied by 10x Genomics refdata-gex-mm10-2020-A Mouse reference, mm10 (GENCODE vM23/Ensembl98). Quality control and analysis were performed with the Seurat package (v4.0.0) in R[66]. Count matrix data from cellranger was imported into a Seurat object using Read10X with the min.cells = 5 & min.features = 200. Data were filtered using thresholds defined during QC analysis (nCount_RNAmin: 1000, nCount_RNAmax: 15,000, nFeature_RNAmin: 200, nFeature_RNAmax: 2500, ribosome_pct>10, percent.mt<5), normalized using the LogNormalize method from Seurat and scaled. PCA analysis was performed using the top 2000 variable features. Doublets were filtered using Doubletfinder (2.0.3)[67], using parameters pN = 0.25, pK = 0.09, nExp=nExp, PCs = 1:10. The number of expected doublets (nExp) was calculated from the theoretical doublet rate multiplied by the number of cells in individual Seurat objects. The merged data was globally normalized using the LogNormalize method from Seurat and scaled. PCA analysis used the top 2000 variable genes. Seurat functions FindNeighbors, FindClusters and RunUMAP were applied with the parameters (dim 1:11, resolution 0.5) for clustering. Differentially expressed genes for each cluster versus the rest were called using Seurat's FindAllMarkers function (applying a non-parametric Wilcoxon rank sum test) with the following pre-filters: min.pct 0.25; 25% of cells in at least one group must express the gene; minimum logfc.threshold of 0.25 between the groups. Plots in Fig. 6a–d were produced using the Seurat functions DimPlot, VlnPlot and FeaturePlot. For the selective analysis of colon cells (as shown in Fig. 7) clustering was performed as described, with the following modifications: FindVariableFeatures (nfeatures = 2500), dims = 1:30 for the Seurat functions FindNeighbors

and RunUMAP. Visualizations and differential gene analysis in Fig. 7a–c, f were done as described above. Gene signature summary scores were produced using the Seurat function AddModuleScore with gene sets derived from co-expression clustering of bulk data (as shown in Fig. 6d) or gene sets listed in Supplementary Table 1 (as shown in Fig. 7c) and 100 control genes. In this function, all analyzed features are binned based on average expression and the control features are randomly selected from each bin. Heatmaps in Supplementary Figs. 6b and 7a were generated using the Seurat function doHeatmap by randomly sampling 2500 cells. For RNA velocity modeling[68] of colon samples (as shown in Fig. 7d), raw reads were reassigned to the genome including introns using kallisto-bustools (kb-python 0.27.2) with the additional parameter --workflow lamanno and kb count (10xv2). UMAP embeddings and cell IDs were exported from the cellranger based Seurat analysis (see above). The cells in the dataset were sorted to just contain cells from the Seurat analysis. Subsequently, scvelo (0.2.4) was applied, with filtering and normalizing of the data, followed by calculation of first and second order moments and velocity estimation in mode: *stochastic*. This was then projected on the UMAP embeddings derived from the Seurat analysis. Processed single-cell sequencing data can be viewed at https://pubdata.lit.eu/treg_proph.

### scTCR-seq analysis

Fastq files were processed using cellranger (version 5.0.0) based on the mm10 reference genome (refdata-cellranger-vdj-GRCm38-alts-ensembl-5.0.0, provided by 10X Genomics). Quality control metrics are provided in Supplementary Table 11. Only cells containing the α/β pair of TCR chains were used for subsequent analyses using the immunarch package (v.0.6.6) in R (v4.0.3). Circos plots presenting clone frequency overlaps (as shown in Fig. 6e) were generated using the circlize R package (0.4.12). Barycentric distributions of clonotype counts (as shown in Fig. 6f) were plotted using ggplot2. Inverse Simpson indices (as shown in Supplementary Fig. 6c) were calculated using the repDiversity function of immunarch (downsampled data, method "inv.simp") and plotted using ggplot2. TCR overlaps between samples were analyzed using the repOverlap function of immunarch with the method "morisita". The clustered heatmap of Morisita's overlap indices shown in Supplementary Fig. 6d and the bar plot of *Trbv* gene usage shown in Supplementary Fig. 6e were generated using ggplot2. TCR clones were visualized or subsets were selected and summarized (as shown in Fig. 6h, i) using the DimPlot and VlnPlot functions from the Seurat package. Counts for individual clones were plotted using ggplot2 (Fig. 6g). Two-sided Fisher exact tests on distributions of *Trbv* sets (Supplementary Figs. 6f and 7e) were performed using the fisher.test function of the stats package (v3.6.2), adjusted for multiple testing using the p.adjust function (stats package) and method BH. Plots were generated using ggplot2.

### Statistics and reproducibility

Statistical analyses were performed in R (v4.0.3/v4.1.0). Analysis methods for flow cytometry or sequencing data are described in the corresponding method sections. All results were independently reproduced. No statistical method was used to predetermine sample size. No data were excluded from the analyses. Animals were randomly allocated to BMT groups. The Investigators were not blinded to allocation during experiments and outcome assessment.

### Reporting summary

Further information on research design is available in the Nature Portfolio Reporting Summary linked to this article.

## Data availability

Raw and processed sequencing data are deposited with the Gene Expression Omnibus (GEO) data repository (GSE223800). The

reference mouse genome assembly (Release M16, GRCm38.p5) was retrieved from Gencode. Source data are provided with this paper.

## Code availability

No novel code was used in this study. Code required to reproduce results and figures are deposited with github (https://github.com/agrehli/Treg-GvHD-prophylaxis).

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

## Acknowledgements
We thank G. Meister and N. Eichner (Department of Biochemistry I, University of Regensburg) for their help in sequencing of bulkTCR libraries. Sequencing of single-cell (sc) RNA-seq libraries was conducted at the biomedical sequencing facility (BSF) of the CeMM (Vienna; Austria). All flow cytometric analysis and cell sorting experiments were performed at the FACS core facility of the Leibniz Institute for Immunotherapy (LIT, Regensburg, Germany). Bulk RNA- and TCR-, as well as scTCR-sequencing were performed at the NGS Core of the LIT (Regensburg, Germany). This study was funded by a grant of the Deutsche Forschungsgemeinschaft to M.E., P.H., and M.R. (Project-ID 324392634—TRR 221, Project B07).

## Author contributions
M.E., P.H. and M.R. designed the study; F.P. performed most animal and FACS experiments with contributions from M.H., E.R., R.E., C.A. and N.W.; D.J.D. performed most NGS experiments with contributions from A.F., H.S., L.S.-P., and C.G.; F.P and P.H. analyzed FACS data; D.J.D., N.S. and M.R. analyzed sequencing data; W.H. provided resources; M.R., P.H. and D.J.D. wrote the original draft with contributions from all authors.

## Funding

## Competing interests
The authors declare no competing interests.
