## [Peer Review File · Nature Communications]

Donor regulatory T cells rapidly adapt to recipient tissues to control murine acute graft-versus-host diseaseREVIEWER COMMENTS

Reviewer #1 (Remarks to the Author):

Summary:

In this manuscript, Dittmar and colleagues provide a comprehensive analysis of organ-specific gene expression profiles of donor-derived Tregs after adoptive transfer in an MHC mismatched mouse model of allo-BMT. Comparing the gene expression profile of donor-derived Tregs re-isolated from spleen, liver, and colon of allo-BMT recipient mice with those of tissue-resident Tregs under steady state, they have shown adaptability of in vitro expanded Tregs to their target organ microenvironment. Authors have shown a striking similarity between the gene expression profile of colon-derived Tregs after adoptive transfer (BMT) and that of colon-resident Tregs under steady state (no BMT). This is an important observation and in keeping with previously published data (PMID 34631716 and PMID 30737144). Activation of hallmark cytotoxic pathways in the colon, when co-transplanted with GVHD-inducing conventional T cells is particularly intriguing and a novel finding. These findings provide valuable insights into the field. The study design is robust and the data are presented in sufficient details and with great clarity. I have a few comments/suggestions as outlined below.

Major comments/suggestions:

- Authors solely rely on the transcriptomic analysis of Tregs to conclude that "Tregs retain their suppressive function and plasticity after transfer." I would encourage authors to include phenotypic and functional data to support these conclusions.
- Authors should compare the gene expression profile of donor Tregs re-isolated from peripheral blood on day 7 post BMT (with/without GVHD) with that of organ-derived Tregs to draw a conclusion about tissue adaptation.

Additional Comments:

Line 42-43: "CD4+CD25+Foxp3+ regulatory T cells (Treg) play a key role in peripheral tolerance and protect from autoimmune as well as infectious disorders:

- Do Tregs protect from infectious disorders? If so, please provide reference to original research. If not, please correct.

Line 124-128: "As shown in Fig. 2b, transferred donor Treg were enriched in the non-lymphoid GvHD target organs colon and liver of allogeneic recipients as compared to physiological levels in non-transplanted mice, while fewer than normal Treg were detectable in recipient spleen and mesenteric lymph nodes seven days after transplantation"

- Absolute number of Tregs (cells per organ) was higher in colon and liver in BMT with adoptive Treg compared to NoBMT without adoptive Treg. Is the increased number of Tregs in GVHD target organs due to "enrichment" in GVHD target organs or simply due to adoptive transfer of Tregs in BMT group (excess Tregs infused)? Need to include a NoBMT with adoptive Treg for comparison of absolute number of Tregs between BMT vs. noBMT groups.

- Likewise, the absolute number of Tregs in the spleen was lower in BMT with adoptive Treg transfer compared to NoBMT without adoptive Treg. Was the lower number of Tregs in the

spleen the effect of BMT (e.g. irradiation) or trafficking of Tregs away from non-GVHD target organs. Comparison with BMT without adoptive Treg is needed to make this conclusion.

Figure 3a: Assume that the group specified as “Donor” in Fig3a and elsewhere (fig 3g and 4d) refers to the donor Tregs after in vitro expansion? Please clarify. Please also include a control “donor” group for Tregs isolated from peripheral blood on day 7 post BMT with/without GVHD for comparison with organ-derived Tregs.

Line 212-217: “Based on expression profiles of CD62L⁻ Treg isolated from non-transplanted C57BL/6 donor mice, we defined similar gene sets distinguishing colon- from liver- and spleen-resident Treg (Supplemental Figs. 3g,h). In line with the UMAP clustering (Fig. 3a), differences between liver- and spleen-resident Treg were small compared to colon – non-colon comparisons”

- How do authors explain similar gene expression between liver and spleen (figure 3a and 3b), even though liver is a GVHD target-organ and spleen is not.
- The process for isolation of Tregs from colon is very different than from liver and spleen. Could the difference in gene expression profile between colon and non-colon (spleen and liver) be in fact due to the differences in isolation process (e.g. cell viability)?

Figure 4d:

- Data presented in Fig 4d shows a similar pattern across spleen, liver, and colon, which are a mix of GVHD-target organs (colon and liver) and non-GVHD target organs (spleen). This raises the question as to whether this “protective gene expression” is specific to tissue-infiltrating tregs and part of tissue adaptation, as implied by the authors, or simply a systemic effect. A comparison between the gene expression profile of peripheral blood vs. organ-derived donor Tregs at day 7 post BMT (w/wo GVHD) is needed to answer this question.

- How do authors explain the lower expression of CD25 (IL2ra) in re-isolated donor-derived Tregs compared to the “donor” group (assume in vitro expanded without BMT)?

Lines 493-495: “This confirms our own previous work on human Treg as well as a report by Hippen et al. demonstrating that Treg retain a stable phenotype and – unlike Tconv – show no signs of exhaustion even after multiple rounds of in vitro stimulation.”

- How does data presented in this manuscript demonstrate that Tregs retain “a stable phenotype and show no signs of exhaustion?”
- Consider referencing previous work by

Line 607-610: From spleen and liver, CD62L⁺ and CD62L⁻ subpopulations were isolated, from colon, only CD62L⁻ cells were obtained (sorting strategy shown in Supplemental Fig.2a). For re-isolation of donor Treg on d7 after transplantation, single cell suspensions from spleen, liver and colon were prepared as described”

- What is the significance of sorting tissue-resident Tregs into CD62L⁺ and CD62L⁻? please explain.
- What percentage of donor derived CD4⁺GFP⁺ Tregs are CD62L⁺ or – after re-isolation from colon?

Reviewer #2 (Remarks to the Author):

Dittmar et al compare the effects of in vitro expanded polyclonal to alloreactive Tregs in their expansion properties, phenotype, transcriptomics and TCR diversity with only the latter substantially diverging, followed by in vivo migration into and organ distribution within GVHD sites. Additional data include comparisons of TCR repertoire between organs and methods of expansion and conclude despite a major overlap, TCR-independent rewiring occurs based upon the organ in which Treg reside. These data are the first that i know of to perform such studies which are important to clinical translation.

Comments-

1. The authors have chosen a day 7 time point for these studies and a single strain combination. The authors should comment as to their thoughts regarding extrapolation of these findings to other strain combinations and why day 7 is an important time to examine. If information were available on other strain combinations or time points that would be helpful to include.
2. Where possible, if not already present, statistical depiction on graphs would be useful (e.g. fig 1b, c; Figure 6i)
3. It would be helpful if the authors synthesized any overall conclusions from figure 3e/f as to the properties of Tregs in colon vs non-colon based upon the annotation provided. Similarly the authors point out differences in figure 3g for clusters 1 vs 3; aside from differences in rewiring, what biological properties may be derived from these differences in these GVHD sites?
4. Figures 5b,c would benefit from additional explanation or examples to aid the reader in understanding the conclusions made for those unfamiliar with barycentric distributions. Additional discussion of Figure 6g, a key point of the manuscript, would also be helpful for the reader.
5. I suggest to add to the discussion the conclusions of a recent paper by Lohmeyer JK et al in Blood that details the TCR repertoire and transcriptome of Tcons as affected by Treg suppression.
6. It would be advantageous for the authors to provide an overall pictorial that summarizes the major findings of the paper.

Reviewer #3 (Remarks to the Author):

In this manuscript the authors compare the phenotypic adaptation of transferred in vitro expanded polyclonal or alloantigen-specific donor Treg in aGvHD. They used an MHC-mismatched mouse model of alloBMT and follow the fate of in vitro expanded donor Tregs. To reach their goal they employed a comprehensive gene expression and repertoire profiling.

The authors showed that Tregs maintain their suppressive function and plasticity after transfer. They observed that upon entering non-lymphoid tissues, and in particular the colon, donor Tregs acquire organ-specific gene expression profiles resembling tissue-resident Tregs. The analysis of the TCR repertoire is confusing for me.

This is an interesting study. I have a few comments that I have listed below.

1. As general comment, it is not clear to me the practical relevance of the observation that injected Tregs adapt in vivo, in particular in the colon where from what I understood the authors do not show any difference between allo- and poly-Tregs. In addition, how this observation can lead to changes in the Treg protocol that has been used so far in GvHD? The authors need to clarify this point.
2. In Figure 2 the authors showed clearly that donor Tregs are superior compared to polyclonal Tregs in protecting from aGvHD and in reducing the Tconv number. However, the distribution of donor and polyclonal Tregs in non-lymphoid organs is not different between the two types of Tregs. Does this observation suggest that the distribution of the two types of cells in the non-lymphoid organs is irrelevant for their function? If this is true we go back to my initial question.
3. The authors then studied the injected Tregs (allo and polyclonal) re-isolated from tissues and they did not observe any difference between the two preparations of Tregs and they decided to combine the two data sets. Why they decided to do so but then they decided again to separate the analysis of the two preparations?
4. Some of the genes that were identified in the re-isolated Tregs in the colon were linked to IFN-I and cytotoxicity and the question is why the authors have not investigated further to understand the relevance of these genes in the injected Treg for their function. In addition, why the authors have not analysed further those genes by separating the allo-specific and polyclonal Tregs? In my view would have added some important information about their different function in protecting from aGvHD?
5. In Figure 5 the authors go back to separating the allo-specific and polyclonal Tregs and show that the infiltration of the Tregs in the organs was mostly driven by alloreactivity. Does this offer an advantage to the allo-Tregs and if this is true why the infiltration of allo-Tregs is not higher than that of the polyclonal Tregs or the percentage of allo-specific Tregs in the polyclonal preparation is lower. Have the authors shown this result?
6. Altogether, I cannot understand why the authors think that their data 'will help guide future approaches to further improve donor Treg-based interventions in aGvHD'. [SEP]

Reviewer #4 (Remarks to the Author):

In this manuscript, David J. Dittmar et al used an MHC-mismatched mouse model of allogeneic bone marrow transplantation to study the fate of in vitro expanded donor Treg upon initial migration to GvHD target organs. It's an important study to explore the functional stability of Treg cells which were adaptive transferred for preventing GVHD in the host.

The authors found that after the transfer, Treg cells retain their suppressive function and plasticity in vivo. In addition, after entering non-lymphoid tissues, donor Treg acquire organ-specific gene expression profiles and showed activated suppressive as well as cytotoxic pathways. Thus, the research reveals donor Treg selection and adaptation mechanisms in GvHD target organs and highlights protective features of Treg in GvHD prevention therapies. But the suppressive function of Treg cells could not be evaluated only by sequencing results, and the inflammatory pathway in Treg cells also affects Treg function and stability. Thus,

there are still several important issues that require the authors' attention, and the manuscript needs to be further improved by fully addressing the following questions and comments:

1. Figure 1 concludes that allogeneic and polyclonal Treg expansion protocols generate cell products with a similar phenotype. But the suppression assay experiment is required to better confirm the suppressive function of allogeneic and polyclonal Treg.
2. In Figure 2, the translation of Treg cells could alleviate transplant-associated complications in the murine MHC-disparate BMT model. The results showed the survival probability and clinical score of mice. But further tests may be required to assess Treg function and inflammation *in vivo*, such as the expression of IL10, IFN γ , and other inflammatory cytokines in the BMT model.
3. Figure 4 showed global gene expression between Treg in GvHD and noGvHD samples across individual organs. And GSEA revealed the significant GvHD-dependent enrichment of a type-I interferon response in spleen and liver-derived Treg. Since the role of interferon in Treg cells infiltrated in different immune environments was not clear, do the results mean that the interferon signaling pathway was essential for Treg cell's function in GVHD samples?
4. In Figure 5, the enrichment of three Trbv segments (Trbv12-1, Trbv12-2, Trbv26) was generally more pronounced in poly than in alloTreg. Trbv12-2 was specific in colon tissue and might respond to microbial, suggesting that the inflammatory microenvironment in the colon may also affect Treg function in GVHD. So investigating the crosstalk of Treg cells and APC cells, T cells, NK cells et al by scRNA-seq may be essential to explore colon Treg development and function in GVHD.

Response to reviewer's comments:

We would like to thank the editor and the reviewers for the time and effort they invested in reviewing our manuscript. In response to their valuable comments and recommendations, we have added new data, including suppression assays and additional FACS data for re-isolated Treg. We have also rewritten large parts of the results & discussion to make it more comprehensible for the broad readership of Nature Communications.

All changes in the main or supplemental text are indicated by red lettering. Changes in figures include:

- Figure 1b: As suggested by one reviewer, we added statistical significance levels.
- Figure 1c: As suggested by one reviewer, we added statistical significance levels.
- Figure 1e: To be more consistent between text and display items, we added labels for additional genes that are further discussed in the main text.
- Figure 5e: We removed a redundant figure to reduce content.
- Figures 4a-c: We removed three GO term annotations and corresponding gene labels which were not further discussed in the main text.
- Figures 3d,f We extended Figure 3f and moved it into the supplement. As suggested by a reviewer, we assigned leading functional annotations to the co-expression clusters shown in Figure 3d.
- Figure S1a As suggested by one reviewer, we added statistical significance levels.
- Figures S1b,c We added new figure panels showing results from suppression assays of in vitro expanded Treg (allo and poly), which was an experiment suggested by a reviewer.
- Figure S2c,d: As suggested, we added additional FACS stains (and corresponding expression data) from re-isolated Treg
- Figure S3c,e: To be more consistent between text and display items, we added gene labels for additional genes showing "memory" in panel S3c that are now also shown in panel S3e.
- Figure S3m: We moved former Figure 3f into the supplement and extended its content. Metascape analyses were complemented with more pathway-oriented EnrichR results to provide a more comprehensive functional annotation.
- Figure S5m: As requested by a reviewer, we added an illustration to explain the design and interpretation of barycentric triangle plots.
- Figure S6c,e: As suggested by one reviewer, we added statistical significance levels.

In addition, we implemented a web-application that will (upon publication) allow everyone to browse through our single cell RNA-seq data. We think that this will greatly improve accessibility to this study. Access is currently password protected, but reviewers may access the web application using the following login data:

https://pubdata.lit.eu/treg_proph
username: review
password: ohPei5

We once again thank all reviewers for their comments and we are convinced that the new data, add-ons and changes significantly improved the manuscript. We hope the reviewers now share our enthusiasm about the novelty and relevance of our findings that significantly advance the research fields of GvHD and Treg therapy, but also provide novel insights into basic Treg biology.

Point-by-point response:

Reviewer #1 (Remarks to the Author):

In this manuscript, Dittmar and colleagues provide a comprehensive analysis of organ-specific gene expression profiles of donor-derived Tregs after adoptive transfer in an MHC mismatched mouse model of allo-BMT. Comparing the gene expression profile of donor-derived Tregs re-isolated from spleen, liver, and colon of allo-BMT recipient mice with those of tissue-resident Tregs under steady state, they have shown adaptability of *in vitro* expanded Tregs to their target organ microenvironment. Authors have shown a striking similarity between the gene expression profile of colon-derived Tregs after adoptive transfer (BMT) and that of colon-resident Tregs under steady state (no BMT). This is an important observation and in keeping with previously published data (). Activation of hallmark cytotoxic pathways in the colon, when co-transplanted with GVHD-inducing conventional T cells is particularly intriguing and a novel finding. These findings provide valuable insights into the field. The study design is robust and the data are presented in sufficient details and with great clarity. I have a few comments/suggestions as outlined below.

We thank reviewer No 1 for the time and efforts in the review of our manuscript and the positive evaluation. We addressed the valuable comments and recommendations and provide a point to point as follows:

Major comments/suggestions:

- Authors solely rely on the transcriptomic analysis of Tregs to conclude that “Tregs retain their suppressive function and plasticity after transfer.” I would encourage authors to include phenotypic and functional data to support these conclusions.

We agree with reviewer #1 that it is important to show that *in vitro* expanded Treg retain their functional activity after transfer into alloBMT recipients. We think the results presented in Fig. 2, namely the suppression of donor Tconv proliferation and expansion in several organs as well as the protection from lethal GvHD and the improvement of clinical manifestations, represent the most important proof of function *in vivo*. Furthermore, we previously showed that *in vitro* expanded Treg show disease-ameliorating activity even when administered after GvHD induction (Riegel et al., 2020). Tregs supported Paneth cell regeneration and improved immune reconstitution, strengthening our notion that *in vitro* expanded Treg retain their functional activity, not only with respect to Tconv suppression, but also their tissue regenerative function. For a more detailed response regarding single suppressive mechanisms and their role in GvHD prevention by Treg we kindly point to our response to reviewer #4. Yet, the focus of this study was to analyze in detail the tissue-specific adaptation of the transplanted donor Treg and the induction of suppressive programs in separate organs in the presence and absence of GvHD-inducing Tconv, which we feel is a new and highly valuable contribution to the field of allogeneic HSCT and GVHD prevention by donor Treg.

With our notion of ‘plasticity’ we wanted to point out that gene expression profiles differ not only between cells of the input Treg graft and those re-isolated from alloBMT recipients 7d post administration, but also between donor Treg reisolated from different tissues due to organ-specific imprinting. Thus, we see a phenotypic and functional adaptation in response to the respective tissue microenvironment. To further support the gene expression data, we now included additional FACS data showing expression of the homing receptors CCR9, CD103 and LPAM-1, the differentiation marker CD62L and the proliferation marker Ki-67 in *in vitro* expanded donor Treg prior to transfer as well as in donor Treg re-isolated from bone marrow, spleen, liver, mesenteric lymph nodes and colon of allogeneic recipients 7d after transfer. In addition, we determined protein expression levels of the tissue Treg marker KLRG-1 on donor Treg after *in vitro* expansion and d7 after transfer (see new Supplemental Fig. 2d & e). These data correlate with the respective mRNA data and clearly confirm a differential, graft- and organ-dependent expression pattern for these markers.

Finally, we would like to refer to the recent study by Lohmeyer et al. (Lohmeyer et al., 2023), which we now also included in our discussion. Using the same alloBMT model (C57BL/6  BALB/c) and also transcriptomic as well as TCR repertoire diversity analysis, the authors demonstrated that Treg, although limiting Tconv proliferation, do not decrease their clonal diversity during suppression in GvHD but rather change the quality of their immune response. The study was limited to pooled cells re-isolated from spleen and LN and focused more on Tconv than Treg analysis. Here we provide a perfectly complementary extension to this study, showing that Treg in different organs and under different inflammatory conditions (GVHD versus no GVHD versus steady-state) upregulate effector molecules to a different degree. For example, our data confirm the observation by Lohmeyer et al. that donor Treg re-isolated from host spleen show a strong upregulation of IL-10 during GvHD suppression as compared to expression in input cells. Interestingly, we could show now that this is less so after alloBMT without induction of GvHD (no Tconv co-transplanted), indicating that the degree of inflammation might indeed regulate gene expression. In colon, a similar shift could be observed between input cells and those retrieved from the organ 7d later; however, in this organ IL-10 expression was already high in steady-state and in fact higher than after alloBMT without GvHD, most probably due to the presence of peripherally induced Treg that are known to express high levels of IL-10. In contrast, donor Treg in liver showed only a minor upregulation of IL-10 even in the presence of Tconv (see Fig. 4d). Thus, the data set for this one gene alone already shows the complex and diversified reaction of Treg upon entering different organs in alloBMT and hopefully convinces reviewer #1 of the importance of our study.

- Authors should compare the gene expression profile of donor Tregs re-isolated from peripheral blood on day 7 post BMT (with/without GVHD) with that of organ-derived Tregs to draw a conclusion about tissue adaptation.

We compare donor Tregs isolated from different organs between each other and with the *in vitro* expanded cells before transplantation. In our view, the comparison of organ-reisolated to input cells clearly illustrates their tissue-specific adaptation. While it may be interesting to add Tregs from blood to our analysis, this would only add an additional tissue that may highlight genes that are commonly changed upon transit from blood to tissue (or vice versa). However, it is not trivial to retrieve sufficient cell numbers from mouse peripheral blood 7 days after alloBMT, especially as we analyzed and compared cells from individual recipients (and organs) throughout our study. The reviewer may also consider that we cannot simply add another population from a different experiment to our analyses. The current experiments were done using a single donor Treg expansion culture for replicate transplantations to be able to compare TCR repertoires across animals. No new experiment would be comparable to previous experiments concerning TCR repertoire analyses since every culture has its own, unique TCR composition. To allow proper batch correction in transcriptome analyses and to study TCR repertoire, we'd essentially have to repeat the Treg isolation also from the other tissues, hence do everything all over again, what would take at least another two years. Since the analysis of PB Treg would not change the central message of our results, we hope the reviewer accepts our lack of PB Treg data.

Additional Comments:

Line 42-43: "CD4+CD25+Foxp3+ regulatory T cells (Treg) play a key role in peripheral tolerance and protect from autoimmune as well as infectious disorders:

- Do Tregs protect from infectious disorders? If so, please provide reference to original research. If not, please correct.

Our initial statement may have been insufficiently explained. Of course, Tregs do not directly contribute to immunity against infectious diseases. Yet, there is ample evidence from several publications by well-known experts that Treg are important for regulating Th1, Th2 and Th17-mediated immune responses to infectious agents both at barrier sites and within the body (Hall et al., 2012; Shafiani et al., 2013; Ting et al., 2018). To do so they upregulate respective

transcription factors and chemokine receptors (e.g. Tbx21 and CXCR3 for suppression of Th1 effector cells), which allows them to home to the respective organs and locally down-regulate inflammatory processes. Thus, they are pivotal for preventing overshooting immune responses to pathogens and for re-establishing homeostasis and thereby protect from infectious diseases and even enable the generation of memory responses (Belkaid et al., 2002). Since our statement seemed to distract, we now omitted this part of the sentence in the introduction, as it is not essentially relevant for the data presented in the manuscript.

Line 124-128: "As shown in Fig. 2b, transferred donor Treg were enriched in the non-lymphoid GvHD target organs colon and liver of allogeneic recipients as compared to physiological levels in non-transplanted mice, while fewer than normal Treg were detectable in recipient spleen and mesenteric lymph nodes seven days after transplantation"

- Absolute number of Tregs (cells per organ) was higher in colon and liver in BMT with adoptive Treg compared to NoBMT without adoptive Treg. Is the increased number of Tregs in GVHD target organs due to "enrichment" in GVHD target organs or simply due to adoptive transfer of Tregs in BMT group (excess Tregs infused)? Need to include a NoBMT with adoptive Treg for comparison of absolute number of Tregs between BMT vs. noBMT groups.

By comparing 'noBMT' i.e. physiological levels of Treg in healthy, 'steady-state' conditions to 'BMT' i.e. Treg cell numbers within particular organs 7d after adoptive transfer, we wanted to provide a reference to make it easier for the reader to contextualize the results. Since we see a rise in Treg cell numbers in non-lymphoid (liver and colon) and a decrease in lymphoid organs (spleen and LN) as compared to steady-state levels, we feel this can neither be solely explained by a higher number of available Treg (due to adoptive transfer), nor by irradiation-induced tissue damage (next reviewer point). We are thus confident that the Treg cell numbers we see in the organs of transplanted mice are the net result of differential homing, retention and activation signals acting on the transferred Treg upon entering recipient organs. The adoptive transfer of Treg into healthy, non-irradiated, T cell-replete animals (no inflammation, niches completely filled) would expose the transferred cells to a completely different situation from that of alloBMT (irradiation-induced inflammation, empty niches). In addition, MHC-mismatched donor cells would rapidly be identified as 'foreign' under these circumstances and eliminated, and transfer of congenic cells would hardly be informative. We thus think that such an approach would not provide relevant information regarding the migratory behavior of donor Treg in the context of alloBMT.

- Likewise, the absolute number of Tregs in the spleen was lower in BMT with adoptive Treg transfer compared to NoBMT without adoptive Treg. Was the lower number of Tregs in the spleen the effect of BMT (e.g. irradiation) or trafficking of Tregs away from non-GVHD target organs. Comparison with BMT without adoptive Treg is needed to make this conclusion.

We also considered this aspect in the design of our experiment. We therefore included the GVHD group ('BMT without Treg'). However, as no Treg were transferred on d0, hardly any donor Treg could be detected in these animals 7 days post BMT, a time point too early to detect donor Treg reconstituted from transplanted stem cells. Thus, a comparison with this group is not informative with respect to Treg trafficking but illustrates the Treg deficiency on d7 post BMT.

Figure 3a: Assume that the group specified as "Donor" in Fig3a and elsewhere (fig 3g and 4d) refers to the donor Tregs after *in vitro* expansion? Please clarify. Please also include a control "donor" group for Tregs isolated from peripheral blood on day 7 post BMT with/without GVHD for comparison with organ-derived Tregs.

We apologize for the lack of explanation regarding the group specified as "Donor". We have now amended the legend of Fig. 3a to: 'UMAP embedding of donor Treg **either after *in-vitro* expansion before transfer ('Donor'), or re-isolated from recipient organs 7d after transfer,...**' and the corresponding manuscript text to: '**We** compared the data sets to those from ***in vitro***

expanded Treg cell products before transfer ("Donor") as well as' to make it clearer that this population represents the expanded donor Treg before transplantation. For reasons stated above, we focus on Treg isolated from organs while data on blood derived Treg cannot be provided and are not expected to change any of our conclusions.

Line 212-217: "Based on expression profiles of CD62L⁻ Treg isolated from non-transplanted C57BL/6 donor mice, we defined similar gene sets distinguishing colon- from liver- and spleen-resident Treg (Supplemental Figs. 3g,h). In line with the UMAP clustering (Fig. 3a), differences between liver- and spleen-resident Treg were small compared to colon – non-colon comparisons"

- How do authors explain similar gene expression between liver and spleen (figure 3a and 3b), even though liver is a GVHD target-organ and spleen is not.

This may have been a misunderstanding, and we apologize if this has not been clearly stated. The quote above refers to differences in resident memory (CD62L⁻) Treg from non-transplanted mice and to our observation that the tissue-specific imprint between liver and spleen is moderate compared to colon in healthy mice. We clarify this misunderstanding now in the revised text to Fig. 3c and Suppl. Figs. 3g-k: 'A similar overlap was observed with gene sets distinguishing colon- from spleen- and liver-resident, memory/effector-type Treg'. In the BMT setting, spleen and liver do separate (Figure 3a and Figure 3e), but the imprint of colon is still much stronger. One reason for the major difference in gene expression between colon and the other two analyzed organs might be the fact that colon is a barrier tissue and thus harbors Treg (both in steady state and after BMT) that are highly specialized in interacting with either commensal bacteria and nutritional metabolites or pathogenic intruders. Whatever the reason, colon seems to be a specialized niche for Treg function and adaptation.

Of note, even though spleen is not considered a classical GvHD target tissue since its affection does not immediately lead to clinical manifestations, it clearly is targeted by GVHD in mouse models including the one used in our study, leading to tissue fibrosis and significant decrease in cellularity (see e.g. Riegel et al, 2020).

- The process for isolation of Tregs from colon is very different than from liver and spleen. Could the difference in gene expression profile between colon and non-colon (spleen and liver) be in fact due to the differences in isolation process (e.g. cell viability)?

We agree with this reviewer that the differences in cell isolation may influence gene signatures. Tissues were isolated using different protocols and the short enzymatic digestion of colon tissues at 37°C may explain a fraction of gene expression differences between colon and the other tissues, based on observations by others (e.g. (Denisenko et al., 2020)). Since we kept incubation times as short as possible and sorted on live cells, we consider the impact on gene expression profiles of individual isolation protocols relatively small compared to the tissue-specific imprint (which has previously also been reported in other studies). In response to this reviewer's legitimate considerations, we have added a sentence describing this possible limitation in the results section (related to Fig. 3b).

Figure 4d:

- Data presented in Fig 4d shows a similar pattern across spleen, liver, and colon, which are a mix of GVHD-target organs (colon and liver) and non-GVHD target organs (spleen). This raises the question as to whether this "protective gene expression" is specific to tissue-infiltrating tregs and part of tissue adaptation, as implied by the authors, or simply a systemic effect. A comparison between the gene expression profile of peripheral blood vs. organ-derived donor Tregs at day 7 post BMT (w/wo GVHD) is needed to answer this question.

Data presented in Fig 4d highlight expression profiles of genes that were previously associated with suppressive mechanisms of Tregs. Since all organs analyzed are infiltrated by donor Tconv and hence in an inflammatory state at the time of analysis, it is plausible that several of these Treg genes are upregulated at all locations. Although similar at first glance, there are

also clear differences between the organs at steady state as well as after BMT without GVHD and with GVHD (as detailed above for IL-10), indicating organ-associated 'programs' that are induced in different situations (steady state vs BMT w/o GVHD vs BMT w GVHD) and different organs, as outlined in the "Discussion" section. In addition, the described genes are just a small (yet important) extraction of all genes analyzed to support the notion that Treg adapt to different tissues after their transfer. The full set of differentially expressed genes is provided as Source Data.

Again, we think that the comparison of input cells to re-extracted donor Treg from separate organs provides at least the same level of scientific information as the comparison to PB Treg.

- How do authors explain the lower expression of CD25 (IL2ra) in re-isolated donor-derived Tregs compared to the "donor" group (assume *in vitro* expanded without BMT)?

During *in vitro* culture Treg are strongly stimulated via TCR and CD28 and exposed to high concentrations of IL-2. Since *Il2ra* expression is also regulated by IL-2 (Kim et al., 2001), this leads to high CD25 expression levels during *in vitro* expansion. Upon transfer into allogeneic hosts, IL-2 levels within organs are presumably lower and thus result in relative downregulation of CD25, which is depicted in the Z score normalized data shown in Fig. 4d. Yet, this form of data presentation only indicates relative differences, and the normalized read counts shown in Suppl. Figure 4d reveal that *Il2ra* expression levels in tissues are still very high.

Lines 493-495: "This confirms our own previous work on human Treg as well as a report by Hippen et al. demonstrating that Treg retain a stable phenotype and – unlike Tconv – show no signs of exhaustion even after multiple rounds of *in vitro* stimulation."

- How does data presented in this manuscript demonstrate that Tregs retain "a stable phenotype and show no signs of exhaustion?"

- Consider referencing previous work by

This comment may also be based on a misunderstanding, and we now try to better describe the context of stability and exhaustion. Regarding stability, we previously showed for naïve human Treg that their TSDR remains demethylated during *in vitro* expansion, thus ensuring stable expression of Foxp3, the lineage-defining transcription factor for Treg (Baron et al., 2007). Here, we show stable protein expression for Foxp3 and Helios after *in vitro* expansion (see Fig. 1d), both key markers for thymus-derived Treg (for this see also our previous work Riegel et al., 2020). In addition, Fig. 1f shows key Treg signature genes identified in several previous studies (see Suppl. table 1 for a complete list) that remain constant during *in vitro* expansion. All these data together (and in combination with functional data after *in vitro* expansion such as suppression assays (see below) and MLRs (see Riegel et. al, 2020) show that Treg retain their key phenotypic markers, and their suppressive function, all of which define them as stable Treg. Regarding exhaustion, we did not mean to refer to a specific gene signature, but simply to the fact that we see Treg proliferating and expanding *in vivo* even after previous *in vitro* expansion, indicating that they are not "exhausted", but still prevent aGvHD. Previous work has defined several "exhaustion" markers in T cells including TIM3 (*Havcr2*), LAG3, PD-1 (*Pdcd1*), GTR (*Tnfrsf18*), TIGIT or TOX. In Treg cells these markers are less well established and so far predominantly studied in CAR-Treg upon tonic signaling (Dong et al., 2019; Lamarche et al., 2023; Shive et al., 2021). Hence, we did not include an analysis of exhaustion signatures in our manuscript. Yet, to inform the reviewers on potential "exhaustion" marker expression levels in expansion cultures, we generated bar plots for the above mentioned markers. As illustrated below, several of the markers showed either only a low expression in freshly isolated as well as *in vitro* expanded Treg (*Havcr2* encoding TIM3) or were not further induced during expansion (*Pdcd1* encoding PD-1 and *Tox*). For the remaining T cell "exhaustion" markers, we saw an increase, particularly in alloTreg cultures, which may be explained by activation rather than exhaustion.

Figure for Review. Expression profiles of putative Treg exhaustion markers. Bars represent mean \pm SE of RPKM values from n=4-6 independent experiments. Individual data points are shown as dots.

Line 607-610: From spleen and liver, CD62L⁺ and CD62L⁻ subpopulations were isolated, from colon, only CD62L⁻ cells were obtained (sorting strategy shown in Supplemental Fig.2a). For re-isolation of donor Treg on d7 after transplantation, single cell suspensions from spleen, liver and colon were prepared as described”

- What is the significance of sorting tissue-resident Tregs into CD62L⁺ and CD62L⁻? please explain.

On mouse lymphocytes, CD62L expression (together with CD44) usually distinguishes naïve from memory/effector cells. Since we started our cultures with naïve (CD62L⁺) donor Treg cells and since Treg (unlike Tconv) retain much of their CD62L expression during *in vitro* expansion (see new Suppl. Fig 2 d & e and also Riegel et al., 2020), but differentially downregulate this marker upon entry into recipient organs, we regarded their comparison to both tissue resident subpopulations (CD62L⁺ and CD62L⁻) as being most informative for the reader. This aspect was obviously not sufficiently described in the initial manuscript and we therefore added a respective explanation in the text related to Fig. 3 and Suppl. Fig 3 in the results section of the manuscript to: “Since Treg cultures were set up with naïve (CD62L⁺) cells that retain CD62L expression during *in vitro* expansion but downregulate the marker after transfer into allogeneic recipients (Supplemental Fig. 2d), naïve and effector/memory (CD62L⁻) Treg populations isolated from the respective organs of non-transplanted donor mice (“no BMT/resident”) were also included in the analysis”.

- What percentage of donor derived CD4⁺GFP⁺ Tregs are CD62L⁺ or – after re-isolation from colon?

Upon isolation from colon, donor Treg show no CD62L expression (see also new Suppl. Fig. 2d & e), neither do resident Treg (hence the isolation of only CD62L⁻ colon Treg from non-transplanted mice, see question above). This is an expected result, as mainly activated (and thus suppressive Treg) seem to migrate to GvHD target organs. We addressed this issue in a new sentence in the legend to Fig. 3a: ‘Unlike spleen and liver, colon harbors only CD62L⁻ Treg.’

Reviewer #2 (Remarks to the Author):

Dittmar et al compare the effects of *in vitro* expanded polyclonal to alloreactive Tregs in their expansion properties, phenotype, transcriptomics and TCR diversity with only the latter substantially diverging, followed by *in vivo* migration into and organ distribution within GVHD sites. Additional data include comparisons of TCR repertoire between organs and methods of expansion and conclude despite a major overlap, TCR-independent rewriting occurs based

upon the organ in which Treg reside. These data are the first that i know of to perform such studies which are important to clinical translation.

We also thank reviewer No 2 for his/her time and efforts in the review of our manuscript and the positive evaluation. We address the valuable comments and recommendations and provide a point-by-point response as follows:

1. The authors have chosen a day 7 time point for these studies and a single strain combination. The authors should comment as to their thoughts regarding extrapolation of these findings to other strain combinations and why day 7 is an important time to examine. If information were available on other strain combinations or time points that would be helpful to include.

Day 7 after transplantation was chosen for analysis based on previous work (Beilhack et al., 2005; Nguyen et al., 2007). Applying *in vivo* BLI, those studies indicated that by this time point donor T cells (Tconv and Treg) show profound expansion in lymphoid organs as well as migration to and substantial seeding of peripheral, non-lymphoid tissues, most importantly the GI tract. Since inflammatory processes and tissue destruction within the GI tract are critically important for disease outcome, we regarded d7 in this complete MHC-mismatch model as particularly informative. We agree that further time points and additional transplant models would be interesting, yet such analyses will be performed in separate sets of experiments (that take a lot of time and effort and are beyond the scope of this manuscript). We think our findings can be extrapolated to other strain combinations as far as MHC (class II) disparities exist, e.g. haplo-identical transplantation, as allo-reactivity (against mismatched MHC) is still expected to provide the strongest signal for prophylactically applied donoTreg. Yet, in MHC-matched, minor antigen mismatched BMT we would expect other precursor frequencies of responsive donor Treg and thus differences in activation, migration and suppressive efficacy. Such experiments are currently in preparation but take time and huge efforts by the whole research team.

2. Where possible, if not already present, statistical depiction on graphs would be useful (e.g. fig 1b, c; Figure 6i)

We thank the reviewer for this important note. We added the statistics to the Figures where it was missing, including Fig. 1b, 1c, Suppl Figure 1a as well as Fig 6i and Suppl Figs 6c & 6e.

3. It would be helpful if the authors synthesized any overall conclusions from figure 3e/f as to the properties of Tregs in colon vs non-colon based upon the annotation provided. Similarly the authors point out differences in figure 3g for clusters 1 vs 3; aside from differences in rewiring, what biological properties may be derived from these differences in these GVHD sites?

As suggested by this reviewer, we tried to simplify the presentation of functional annotation of co-expression clusters and have removed the GO term enrichment (original Fig. 3f) to the supplement. To add some more meaningful annotation to each cluster we performed additional pathway enrichment analyses and now provide lead functional annotation terms/pathways to each co-expression cluster (revised Figure 3d and in more detail in Suppl. Fig. 3m). We profoundly rephrased the description of the results related to Fig 3d and now highlight prominent functional annotations that e.g. distinguish cluster 1 from cluster 3.

4. Figures 5b,c would benefit from additional explanation or examples to aid the reader in understanding the conclusions made for those unfamiliar with barycentric distributions. Additional discussion of Figure 6g, a key point of the manuscript, would also be helpful for the reader.

We originally provided some explanation in the Figure 5 legend, but obviously failed to thoroughly describe this rather new type of data presentation. To make the graphs more comprehensible, we added a schematic in Suppl. Figure 5m that explains the meaning of bubble size and position, as well as providing a guide for the interpretation of these triangle plots.

Regarding Figure 6g, we have now also amended the description of this part in the text to make the data more comprehensible. We hope that our amendments and alterations in the text improved the comprehension of these rather detailed and large figures.

5. I suggest to add to the discussion the conclusions of a recent paper by Lohmeyer JK et al in Blood that details the TCR repertoire and transcriptome of Tcons as affected by Treg suppression.

We thank the reviewer for this valuable suggestion and have now added a paragraph to the discussion in which we highlight the complementarity of the two studies. Our analyses perfectly complement the experiments of the Negrin Lab (Lohmeyer et al., 2023) on pooled donor Treg re-isolated from recipient spleen and lymph nodes and extend them to the analysis of transferred Treg retrieved from colon and liver, that was not performed in their study (see also our response to reviewer #1 on this matter).

6. It would be advantageous for the authors to provide an overall pictorial that summarizes the major findings of the paper.

We thank the reviewer for this suggestion and now provide a graphical summary of our findings in Figure 8.

Reviewer #3 (Remarks to the Author):

In this manuscript the authors compare the phenotypic adaptation of transferred *in vitro* expanded polyclonal or alloantigen-specific donor Treg in aGvHD. They used an MHC-mismatched mouse model of alloBMT and follow the fate of *in vitro* expanded donor Tregs. To reach their goal they employed a comprehensive gene expression and repertoire profiling. The authors showed that Tregs maintain their suppressive function and plasticity after transfer. They observed that upon entering non-lymphoid tissues, and in particular the colon, donor Tregs acquire organ-specific gene expression profiles resembling tissue-resident Tregs. The analysis of the TCR repertoire is confusing for me. This is an interesting study. I have a few comments that I have listed below.

We thank reviewer No 3 for his/her valuable time and effort dedicated to the review of our manuscript. We address the important comments and recommendations and provide a point-by-point response as follows:

1. As general comment, it is not clear to me the practical relevance of the observation that injected Tregs adapt *in vivo*, in particular in the colon where from what I understood the authors do not show any difference between allo- and poly-Tregs. In addition, how this observation can lead to changes in the Treg protocol that has been used so far in GvHD? The authors need to clarify this point.

We apologize for not sufficiently describing the rationale for our experiments in the text of the manuscript. We modified various paragraphs, particularly in the results section, to better describe our research aims.

The primary goal of our investigation was to determine the plasticity and functional flexibility of *in vitro* expanded donor Treg after transfer into allogeneic hosts. To our knowledge, this is the first report demonstrating that repeated stimulation and substantial expansion *in vitro* prior to

transplantation does not lead to a 'locked in' state of differentiated donor Treg. In contrast, our data show that they migrate to different tissues and react to local stimuli in a highly flexible and adaptive way. This finding is of importance not only for their use in GvHD, but also with respect to applications in other diseases and, most importantly, for the elucidation of general Treg biology.

Since Treg represent only a minor population of CD4⁺ T cells in PB in humans (3-8%), their isolation in sufficient numbers is challenging. Thus, Treg *in vitro* expansion seems necessary for many clinical applications. We were among the first to describe efficient *in vitro* expansion protocols for human Treg (Hoffmann et al., 2004) and translated them to the clinics (EudraCT #2012-002685-12 and 2016-003947-12). So far, we investigated and applied polyclonally expanded donor Treg. Since other publications suggested that host-directed donor-Treg are more potent in suppressing GVHD than polyclonally expanded ones (w/o describing underlying mechanisms), we now compared these strategies. Indeed, alloTreg were more effective in our complete-MHC-mismatched BMT model, even though they did not significantly differ to polyTreg with respect to their migration patterns, their organ-specific adaptation processes nor their upregulation of suppressive mechanisms. Based on these findings we conclude that the frequency of host reactive donor Treg in the graft is most important for early GvHD inhibition in this model. In other donor-host constellations (e.g. less or no MHC discrepancy), this may be different and polyclonal Treg may even be advantageous. We further explore this in ongoing experiments. Yet, we are convinced that the provided data are highly relevant for the elucidation of basic Treg biology, but also for clinical translation, e.g. for novel strategies aiming at TCR- or CAR-modification of donor Treg.

2. In Figure 2 the authors showed clearly that donor Tregs are superior compared to polyclonal Tregs in protecting from aGvHD and in reducing the Tconv number. However, the distribution of donor and polyclonal Tregs in non-lymphoid organs is not different between the two types of Tregs. Does this observation suggest that the distribution of the two types of cells in the non-lymphoid organs is irrelevant for their function? If this is true we go back to my initial question.

In fact, we see a comparable abundance of allo and polyTreg in the analyzed organs on d7 after transplantation. Yet, it is important to remember that there are two distinct mechanisms of T cell expansion after alloBMT. In lethally irradiated hosts, donor T cells exhibit homeostatic proliferation (due to the empty niches) as well as antigen-specific responses, particularly towards MHC-disparities. Allo-specific responses seem pivotal for the suppression of aGvHD, as Treg only initiate their suppressive machinery upon TCR-stimulation. The higher precursor frequency for ubiquitously expressed allo-MHC in *in vitro* primed alloTreg results in the more rapid and profound suppression of Tconv expansion, which translates into better overall survival. Even though allo and poly Treg were equally suppressive *in vitro* (see new Suppl. Fig. 1 b&c), they differ with respect to their TCR repertoire. The more profound effect of alloTreg is thus explained by their higher precursor frequency of allo-reactive clones. This is supported by the fact that re-isolated alloTreg were enriched for clones already expanded during culture (Fig. 5d). Taken together, we think that allo-reactivity is relevant at the Treg priming sites and in GvHD target organs.

3. The authors then studied the injected Tregs (allo and polyclonal) re-isolated from tissues and they did not observe any difference between the two preparations of Tregs and they decided to combine the two data sets. Why they decided to do so but then they decided again to separate the analysis of the two preparations?

We apologize for the confusion and have now added additional text to explain the rationale for combining the allo and poly samples for some gene expression analyses. In essence, we only combined the data sets for the comparative statistical analyses of organ-derived Tregs from individual treatments to add statistical power. However, when we depict expression data of individual genes, we always separate poly and alloTreg, because it is actually informative to keep this separation. The co-expression analysis (in Fig. 3d-f) is done on the basis of individual

samples and basically agnostic to sample types, hence we did not combine allo and poly here either.

4. Some of the genes that were identified in the re-isolated Tregs in the colon were linked to IFN-I and cytotoxicity and the question is why the authors have not investigated further to understand the relevance of these genes in the injected Treg for their function. In addition, why the authors have not analysed further those genes by separating the allo-specific and polyclonal Tregs? In my view would have added some important information about their different function in protecting from aGvHD?

In our experiments, we observed a significant GvHD-dependent upregulation of type 1 interferon response genes in donor Treg re-isolated from recipient spleen and liver, but not in those retrieved from colon (see scatter plots in Fig. 4a-c as well as GSEA in Suppl. Fig. 4a). Notably, those genes did not include type I interferons themselves, suggesting that they represent a consequence of paracrine signaling in an organ-specific inflammatory environment. Genes encoding cytotoxic molecules such as *Gzma/b* and *Prf1*, were significantly upregulated in all three organs in the context of GvHD. Upon separate analysis of allo and polyTreg (Fig. 4d and Suppl. Fig. 4d), we observed only discrete differences between the two Treg populations. Much more strikingly, however, our analyses revealed not only different base levels for these genes in the three organs under steady-state conditions, but also a substantial, yet differential upregulation after BMT depending on the organ and the severity of inflammation (i.e. BMT with vs w/o GvHD). The contribution of various mechanisms to Treg-mediated GvHD suppression are controversially discussed in several previous studies by us and others (for more details in this matter we kindly refer to our response to reviewer #4). The focus of our study was to determine the plasticity and functional flexibility of *in vitro* expanded Treg after transfer into allogeneic hosts. Our work clearly suggests organ and context specific mechanisms of action for donor Treg, which requires further investigation and characterization. The dissection of each individual gene/mechanism contributing to GvHD suppression in individual organs is beyond the scope of this manuscript but opens the avenue for a multitude of future studies.

5. In Figure 5 the authors go back to separating the allo-specific and polyclonal Tregs and show that the infiltration of the Tregs in the organs was mostly driven by alloreactivity. Does this offer an advantage to the allo-Tregs and if this is true why the infiltration of allo-Tregs is not higher than that of the polyclonal Tregs or the percentage of allo-specific Tregs in the polyclonal preparation is lower. Have the authors shown this result?

Q5 seems a variation of Q1-4 and we thus refer to the previous answers. As outlined previously, we significantly modified the text to describe research goals and results more precisely. We hope our modifications improved the manuscript and make it more easily comprehensible to the readers of Nat. Commun.

6. Altogether, I cannot understand why the authors think that their data 'will help guide future approaches to further improve donor Treg-based interventions in aGvHD'.

We refer to our answers to Q1 and Q2 on this topic. We hope the reviewer agrees that the elucidation of Treg function in BMT/HSCT is legitimate and scientifically relevant. Studies like ours aim to improve the knowledge on donor Treg function after allogeneic transplantation. Overall, this information will help to develop improved Treg-based therapeutic interventions, hopefully for the benefit of allo-HSCT patients.

Reviewer #4 (Remarks to the Author):

In this manuscript, David J. Dittmar et al used an MHC-mismatched mouse model of allogeneic bone marrow transplantation to study the fate of *in vitro* expanded donor Treg upon initial migration to GvHD target organs. It's an important study to explore the functional stability of Treg cells which were adaptive transferred for preventing GVHD in the host.

The authors found that after the transfer, Treg cells retain their suppressive function and plasticity *in vivo*. In addition, after entering non-lymphoid tissues, donor Treg acquire organ-specific gene expression profiles and showed activated suppressive as well as cytotoxic pathways. Thus, the research reveals donor Treg selection and adaptation mechanisms in GvHD target organs and highlights protective features of Treg in GvHD prevention therapies. But the suppressive function of Treg cells could not be evaluated only by sequencing results, and the inflammatory pathway in Treg cells also affects Treg function and stability. Thus, there are still several important issues that require the authors' attention, and the manuscript needs to be further improved by fully addressing the following questions and comments:

We thank reviewer No 4 for his/her time and efforts that went into the review of our work and the valuable comments and recommendations that helped us improve our manuscript. We address the important comments and recommendations and provide a point-by-point response as follows:

1. Figure 1 concludes that allogeneic and polyclonal Treg expansion protocols generate cell products with a similar phenotype. But the suppression assay experiment is required to better confirm the suppressive function of allogeneic and polyclonal Treg.

We thank the reviewer for this important suggestion. We have now included the detailed results from *in vitro* suppression assays performed with cells from four individual cultures. These data show that alloTreg and polyTreg do not differ in their suppressive activity prior to transfer. We have included these additional results as new Supplemental Fig. 2b and c.

2. In Figure 2, the translation of Treg cells could alleviate transplant-associated complications in the murine MHC-disparate BMT model. The results showed the survival probability and clinical score of mice. But further tests may be required to assess Treg function and inflammation *in vivo*, such as the expression of IL10, IFN γ , and other inflammatory cytokines in the BMT model.

We agree with the reviewer that it is important to determine the contribution of the various Treg-mediated suppressive mechanisms to GvHD prevention. Several of these mechanisms have been reported by us and others in previous publications. For example, we demonstrated in our very first Treg paper that IL-10 production by donor Treg is an important mechanism for Treg-mediated protection from lethal GvHD (Hoffmann et al., 2002). Bolton et al. (Bolton et al., 2015) described the important role of CTLA-4 expression by Treg in the immediate early phase after transplantation and Yolcu et al., (Yolcu et al., 2013) and more recently Shrestha and colleagues (Shrestha et al., 2023) demonstrated the increased protection from GvHD by Treg transiently equipped with FasL. Similarly, Liu et al. have shown the importance of an autocrine cycle of IL-35 secreted by Treg that increases not only their own suppressive activity but also supports the induction of pTreg and thereby improves protection from GvHD (Liu et al., 2015). Other mechanisms are more controversially discussed, including the involvement of granzymes or perforin. (see e.g. (Gondek et al., 2008)). With regard to inflammation and suppression of pro-inflammatory responses by Treg, we have previously shown in two different models the reduction of IFN- γ and TNF in the serum and in the colon of Treg-treated as compared to non-treated alloBMT recipients (Edinger et al., 2003; Riegel et al., 2020). Taken together, Treg seem to engage several modes of suppression in GvHD and we did not repeat previously published experiments. The novelty and most surprising finding in our current manuscript is indeed the proof that Tregs engage different suppressive pathways to a different

extent within different organs and in response to inflammatory triggers (e.g. in the presence or absence of Tconv (GvHD)). To our knowledge, this is the most detailed and comprehensive coverage of Treg-mediated protection from GvHD so far. These findings now permit the future examination of suppressive mechanisms in an organ and context-specific manner.

3. Figure 4 showed global gene expression between Treg in GvHD and noGvHD samples across individual organs. And GSEA revealed the significant GvHD-dependent enrichment of a type-I interferon response in spleen and liver-derived Treg. Since the role of interferon in Treg cells infiltrated in different immune environments was not clear, do the results mean that the interferon signaling pathway was essential for Treg cell's function in GVHD samples?

The role of type-I interferons in Treg function is indeed not well understood. We see that IFN-response genes are up-regulated in spleen and liver Treg after BMT in the presence of Tconv, i.e. in GvHD. This indicates that the alloreactive environment may induce higher levels of type-I interferons, likely due to the stronger inflammatory reaction in the presence of donor Tconv. While type-I-interferons are generally thought to be protective in GvHD (Robb et al., 2011) we cannot delineate from our data whether interferon signalling is essential for the function of donor Treg. To address this interesting question, additional studies are required, which are beyond the scope of this manuscript.

4. In Figure 5, the enrichment of three Trbv segments (Trbv12-1, Trbv12-2, Trbv26) was generally more pronounced in poly than in alloTreg. Trbv12-2 was specific in colon tissue and might respond to microbial, suggesting that the inflammatory microenvironment in the colon may also affect Treg function in GVHD. So investigating the crosstalk of Treg cells and APC cells, T cells, NK cells et al by scRNA-seq may be essential to explore colon Treg development and function in GVHD.

We agree with this reviewer that it is important to consider the local microenvironment as well as other Treg-interacting cell types in future studies. Here, we primarily focused on donor Treg to characterize their selection and fate in aGvHD. This was essential to identify tissue-specific responses of Tregs and differences in TCR selection. Follow-up studies will address the important interactions between Treg and other cell types within the local GvHD environment and further elucidate the role of Tregs expressing TCRs with the above mentioned Trbv segments. For this purpose, we set up a germ-free mouse facility to perform BMT in germ-free recipients. Yet, these experiments have just started and will be reported once informative data are generated.

References:

Baron, U., Floess, S., Wiczorek, G., Baumann, K., Grutzkau, A., Dong, J., Thiel, A., Boeld, T.J., Hoffmann, P., Edinger, M., Turbachova, I., Hamann, A., Olek, S., and Huehn, J. (2007). DNA demethylation in the human FOXP3 locus discriminates regulatory T cells from activated FOXP3(+) conventional T cells. *Eur J Immunol* 37, 2378-2389.

Beilhack, A., Schulz, S., Baker, J., Beilhack, G.F., Wieland, C.B., Herman, E.I., Baker, E.M., Cao, Y.-A., Contag, C.H., and Negrin, R.S. (2005). In vivo analyses of early events in acute graft-versus-host disease reveal sequential infiltration of T-cell subsets. *Blood* 106, 1113-1122.

Belkaid, Y., Piccirillo, C.A., Mendez, S., Shevach, E.M., and Sacks, D.L. (2002). CD4+CD25+ regulatory T cells control *Leishmania major* persistence and immunity. *Nature* 420, 502-507.

Bolton, H.A., Zhu, E., Terry, A.M., Guy, T.V., Koh, W.P., Tan, S.Y., Power, C.A., Bertolino, P., Lahl, K., Sparwasser, T., Shklovskaya, E., and Fazekas de St Groth, B. (2015). Selective Treg reconstitution during lymphopenia normalizes DC costimulation and prevents graft-versus-host disease. *J Clin Invest* 125, 3627-3641.

Denisenko, E., Guo, B.B., Jones, M., Hou, R., de Kock, L., Lassmann, T., Poppe, D., Clement, O., Simmons, R.K., Lister, R., and Forrest, A.R.R. (2020). Systematic assessment of tissue dissociation and storage biases in single-cell and single-nucleus RNA-seq workflows. *Genome Biol* 21, 130.

Dong, Y., Li, X., Zhang, L., Zhu, Q., Chen, C., Bao, J., and Chen, Y. (2019). CD4(+) T cell exhaustion revealed by high PD-1 and LAG-3 expression and the loss of helper T cell function in chronic hepatitis B. *BMC Immunol* 20, 27.

Edinger, M., Hoffmann, P., Ermann, J., Drago, K., Fathman, C.G., Strober, S., and Negrin, R.S. (2003). CD4+CD25+ regulatory T cells preserve graft-versus-tumor activity while inhibiting graft-versus-host disease after bone marrow transplantation. *Nature Medicine* 9, 1144-1150.

Gondek, D.C., Devries, V., Nowak, E.C., Lu, L.F., Bennett, K.A., Scott, Z.A., and Noelle, R.J. (2008). Transplantation survival is maintained by granzyme B+ regulatory cells and adaptive regulatory T cells. *J Immunol* 181, 4752-4760.

Hall, A.O., Beiting, D.P., Tato, C., John, B., Oldenhove, G., Lombana, C.G., Pritchard, G.H., Silver, J.S., Bouladoux, N., Stumhofer, J.S., Harris, T.H., Grainger, J., Wojno, E.D., Wagage, S., Roos, D.S., Scott, P., Turka, L.A., Cherry, S., Reiner, S.L., Cua, D., Belkaid, Y., Elloso, M.M., and Hunter, C.A. (2012). The cytokines interleukin 27 and interferon-gamma promote distinct Treg cell populations required to limit infection-induced pathology. *Immunity* 37, 511-523.

Hoffmann, P., Eder, R., Kunz-Schughart, L.A., Andreesen, R., and Edinger, M. (2004). Large-scale in vitro expansion of polyclonal human CD4(+)/CD25high regulatory T cells. *Blood* 104, 895-903.

Hoffmann, P., Ermann, J., Edinger, M., Fathman, C.G., and Strober, S. (2002). Donor-type CD4+CD25+ Regulatory T Cells Suppress Lethal Acute Graft-Versus-Host Disease after Allogeneic Bone Marrow Transplantation. *Journal of Experimental Medicine* 196, 389-399.

Kim, H.P., Kelly, J., and Leonard, W.J. (2001). The basis for IL-2-induced IL-2 receptor alpha chain gene regulation: importance of two widely separated IL-2 response elements. *Immunity* 15, 159-172.

Lamarche, C., Ward-Hartstonge, K., Mi, T., Lin, D.T.S., Huang, Q., Brown, A., Edwards, K., Novakovsky, G.E., Qi, C.N., Kobor, M.S., Zebly, C.C., Weber, E.W., Mackall, C.L., and Levings, M.K. (2023). Tonic-signaling chimeric antigen receptors drive human regulatory T cell exhaustion. *Proc Natl Acad Sci U S A* 120, e2219086120.

Liu, Y., Wu, Y., Wang, Y., Cai, Y., Hu, B., Bao, G., Fang, H., Zhao, L., Ma, S., Cheng, Q., Song, Y., Liu, Y., Zhu, Z., Chang, H., Yu, X., Sun, A., Zhang, Y., Vignali, D.A., Wu, D., and Liu, H. (2015). IL-35 mitigates murine acute graft-versus-host disease with retention of graft-versus-leukemia effects. *Leukemia* 29, 939-946.

Lohmeyer, J.K., Hirai, T., Turkoz, M., Buhler, S., Lopes Ramos, T., Kohler, N., Baker, J., Melotti, A., Wagner, I., Pradier, A., Wang, S., Ji, X., Becattini, S., Villard, J., Merkler, D., Chalandon, Y., Negrin, R.S., and Simonetta, F. (2023). Analysis of the T-cell repertoire and transcriptome identifies mechanisms of regulatory T-cell suppression of GVHD. *Blood* 141, 1755-1767.

Nguyen, V.H., Zeiser, R., Dasilva, D.L., Chang, D.S., Beilhack, A., Contag, C.H., and Negrin, R.S. (2007). In vivo dynamics of regulatory T-cell trafficking and survival predict effective strategies to control graft-versus-host disease following allogeneic transplantation. *Blood* 109, 2649-2656.

Riegel, C., Boeld, T.J., Doser, K., Huber, E., Hoffmann, P., and Edinger, M. (2020). Efficient treatment of murine acute GvHD by in vitro expanded donor regulatory T cells. *Leukemia* 34, 895-908.

Robb, R.J., Kreijveld, E., Kuns, R.D., Wilson, Y.A., Olver, S.D., Don, A.L., Raffelt, N.C., De Weerd, N.A., Lineburg, K.E., Varelias, A., Markey, K.A., Koyama, M., Clouston, A.D., Hertzog, P.J., Macdonald, K.P., and Hill, G.R. (2011). Type I-IFNs control GVHD and GVL responses after transplantation. *Blood* 118, 3399-3409.

Shafiani, S., Dinh, C., Ertelt, J.M., Moguche, A.O., Siddiqui, I., Smigielski, K.S., Sharma, P., Campbell, D.J., Way, S.S., and Urdahl, K.B. (2013). Pathogen-specific Treg cells expand early during mycobacterium tuberculosis infection but are later eliminated in response to Interleukin-12. *Immunity* 38, 1261-1270.

Shive, C.L., Freeman, M.L., Younes, S.A., Kowal, C.M., Canaday, D.H., Rodriguez, B., Lederman, M.M., and Anthony, D.D. (2021). Markers of T Cell Exhaustion and Senescence and Their Relationship to Plasma TGF-beta Levels in Treated HIV+ Immune Non-responders. *Front Immunol* 12, 638010.

Shrestha, P., Turan, A., Batra, L., Gulen, A.E., Sun, Z., Tan, H., Askenasy, N., Shirwan, H., and Yolcu, E.S. (2023). Engineering donor lymphocytes with Fas ligand protein effectively prevents acute graft-versus-host disease. *Blood Adv* 7, 2181-2195.

Ting, H.A., de Almeida Nagata, D., Rasky, A.J., Malinczak, C.A., Maillard, I.P., Schaller, M.A., and Lukacs, N.W. (2018). Notch ligand Delta-like 4 induces epigenetic regulation of Treg cell differentiation and function in viral infection. *Mucosal Immunol* 11, 1524-1536.

Yolcu, E.S., Kaminitz, A., Mizrahi, K., Ash, S., Yaniv, I., Stein, J., Shirwan, H., and Askenasy, N. (2013). Immunomodulation with donor regulatory T cells armed with Fas-ligand alleviates graft-versus-host disease. *Exp Hematol* 41, 903-911.

REVIEWER COMMENTS

Reviewer #2 (Remarks to the Author):

i have carefully read the responses to my comments and i find them entirely satisfactory. I have no further requests

Reviewer #3 (Remarks to the Author):

The additional results have answered most of my questions. Altogether, I am satisfied with the answers of the authors.

Editorial Note: This reviewer was additionally asked to comment in place of reviewer 1 who was unavailable to provide a comment during this round. Please see the attachment.

Comments on the answers to the Reviewer 1 questions:

1. The first question of the reviewer was to do some additional experiments to address the function and plasticity of Tregs after transfer. Although the authors have not done what asked regarding the function of the Tregs in vivo after transfer their answer to this first question is satisfactory in my view. Plasticity for the authors is equal to adaptation and they have new data further confirming that the Tregs injected adapt to the tissue/organ where they locate.
2. Then the reviewer asked to include in the Treg analysis of the tissue/organ after injection the analysis of the Tregs in the blood. The authors did not do the experiments. In my view the analysis of blood Tregs after injection does not add much to the question that the authors have addressed namely the adaptation of the Tregs in the different organs and the comparison between the different organs is sufficient.
3. The reviewer requested in the third question about having a control group of mice by including a noBMT cohort with adoptive transfer of Tregs. The authors answered that it was not necessary because they said by injecting the Tregs in a non-irradiated mouse the Tregs will be eliminated. I agree with that and the only alternative group would have been injecting the Tregs in an irradiated animal. The problem there is that there will be homeostatic proliferation of the Tregs that will be different compared to the animals that received a BM and Treg (although I do not have direct experience with this type of animal model). In conclusion, I do not know whether a control group noBMT can be the correct control; however, I agree with the authors that what the reviewer suggested would be 'a completely different situation from that of alloBMT'
4. The next question is linked to the previous question. Namely the need for a control group to understand why the absolute number of Tregs in the spleen was lower in BMT with adoptive transfer compared to nBMT. In this case I think the authors should add the data from the experiments that they have done with mice irradiated and BMT but not Tregs. At least this can clarify whether the reduced number of Tregs in the spleen observed was due to with the irradiation of the mice (as commented by the authors themselves).

5. The authors have addressed the next two questions of the reviewer and Figure legend and text have been modified and clarified.

6. The reviewer raised the possibility that the differences in the gene signature is linked to the way that the cells were isolated from the colon compared to the other organs. The authors have addressed this point and answered but I do not have experience with the isolation of cells from the different organs and I cannot judge whether what the authors answered is good enough.

7. In the next question the reviewer asked again about a comparison with the blood Tregs. The reviewer raises the question whether the change in the gene profile is due to tissue adaptation or systemic effect. I do not understand this question, as the gene profile is different in the different organs. The authors answered by repeating again that there are differences between the organs and that the analysis of the Tregs in the blood will not add very much. I think I agree with them.

8. The next is about the levels of the IL-2R between the expanded Tregs and the Tregs that were re-isolated. The authors provided a convincing answer in my view.

9. The next question is about stable phenotype and exhaustion. The authors replied convincingly on the stable phenotype and for the exhaustion they provide in the rebuttal letter additional data and I agree with them that the markers that characterize exhausted Tconv are not well characterized for Tregs.

10. The authors have addressed the last two questions about CD62L from the reviewer by adding sentences that clarify their statements.

Reviewer #4 (Remarks to the Author):

All the questions have been fully addressed, so I agree that the revised version can be accepted for publication without further delay.

Response to reviewer's comments:

We thank the editor and the reviewers again for their time and effort in reviewing our manuscript. We are specifically grateful to reviewer 3 for his/her additional evaluation of our responses to the issues originally raised by reviewer 1.

We are very pleased to see that reviewers 2, 3 and 4 were convinced by our response and approved the revisions. The remaining two points initially raised by reviewer 1 are addressed in the point-by-point reply below.

Point-by-point response:

Reviewer #2 (Remarks to the Author):

i have carefully read the responses to my comments and i find them entirely satisfactory. I have no further requests

Reviewer #3 (Remarks to the Author):

The additional results have answered most of my questions. Altogether, I am satisfied with the answers of the authors.

Editorial Note: This reviewer was additionally asked to comment in place of reviewer 1 who was unavailable to provide a comment during this round. Please see the attachment.

Comments on the answers to the Reviewer 1 questions:

1. The first question of the reviewer was to do some additional experiments to address the function and plasticity of Tregs after transfer. Although the authors have not done what asked regarding the function of the Tregs in vivo after transfer their answer to this first question is satisfactory in my view. Plasticity for the authors is equal to adaptation and they have new data further confirming that the Tregs injected adapt to the tissue/organ where they locate.

2. Then the reviewer asked to include in the Treg analysis of the tissue/organ after injection the analysis of the Tregs in the blood. The authors did not do the experiments. In my view the analysis of blood Tregs after injection does not add much to the question that the authors have addressed namely the adaptation of the Tregs in the different organs and the comparison between the different organs is sufficient.

3. The reviewer requested in the third question about having a control group of mice by including a noBMT cohort with adoptive transfer of Tregs. The authors answered that it was not necessary because they said by injecting the Tregs in a non-irradiated mouse the Tregs will be eliminated. I agree with that and the only alternative group would have been injecting the Tregs in an irradiated animal. The problem there is that there will be homeostatic proliferation of the Tregs that will differ compared to the animals that received a BM and Treg

(although I do not have direct experience with this type of animal model). In conclusion, I do not know whether a control group noBMT can be the correct control; however, I agree with the authors that what the reviewer suggested would be 'a completely different situation from that of alloBMT'

4. The next question is linked to the previous question. Namely the need for a control group to understand why the absolute number of Tregs in the spleen was lower in BMT with adoptive

transfer compared to nBMT. In this case I think the authors should add the data from the experiments that they have done with mice irradiated and BMT but not Tregs. At least this can clarify whether the reduced number of Tregs in the spleen observed was do to with the irradiation of the mice (as commented by the authors them-self).

In MHC-mismatched GvHD models, donor T cells engraft and expand rapidly in lymphoid tissues and migrate to GvHD target tissues on d5, while the total pool of alloreactive T cells in lymphoid tissues retracts at this time point due to activation-induced cell death and GvHD-induced tissue destruction that ultimately causes fibrosis in lymph nodes and spleen (Beilhack et al., 2005). It is thus not surprising that total cell numbers in lymphoid tissues are lower in BMT recipients as compared to non-transplanted mice. To compare the effect of co-transplanted donor Treg we included the "GvHD group", namely recipients that received BM, conventional donor T cells, but **no** donor Treg. Thus, this is the requested control group where the combined effects of irradiation and GvHD on organ cellularity can be compared. In these mice, only few donor-BM-derived or descendants of residual Treg from the Tconv cell product (CD25-depleted) or induced Treg (from Tconv) are detected. Please find the complete data set in Fig. 1 of this response letter. Nevertheless, we'd prefer not to include these data in the final manuscript as they distract from our research topic. The main topic is the fate and differentiation of *in vitro* expanded donor Treg after adoptive transfer and their influence on GvHD-inducing conventional T cells. The few residual or induced Treg in BMT recipients are functionally irrelevant as evident by the clinical outcome of the animals in the GvHD group (BMT, Tconv but no Treg) and thus not important for the scope of our experiments. Taken together, we think that the relevant controls are shown in the manuscript. The "no BMT" group is solely shown to illustrate Treg effects in GvHD prevention in relation to physiological Treg levels in non-transplanted mice.

Figure 1 for Review: **Absolute cell numbers of CD4+Foxp3+ Treg cells derived from transplanted donor BM/Tconv cells or additionally transferred donor allo/poly Treg cells.** BALB/c recipients were lethally irradiated and received donor (B6_CD45.1⁺) TCD BM and Tconv cells either alone (BMT w/o Treg = GVHD group) or together with *in vitro* expanded donor allo or poly Treg from B6_CD45.2⁺ Foxp3^{gfp} reporter mice. Mice were sacrificed 7d later, organs were retrieved and TCD BM/Tconv- as well as allo/polyTreg-derived Treg cells were determined. Bar plots show absolute cell numbers per organ of BM/Tconv-derived Treg (gray bars; CD45.1⁺CD4⁺Foxp3⁺/gfp⁻ T cells) as well as allo or poly Treg-derived cells (orange and blue bars, respectively; CD45.2⁺CD4⁺Foxp3⁺/gfp⁺ T cells). Bars represent mean values ± SE (log scale) and dots represent data from individual experiments.

5. The authors have addressed the next two questions of the reviewer and Figure legend and text have been modified and clarified.

6. The reviewer raised the possibility that the differences in the gene signature is linked to the way that the cells were isolated from the colon compared to the other organs. The authors have addressed this point and answered but I do not have experience with the isolation of cells from the different organs and I cannot judge whether what the authors answered is good enough.

Reviewer 1 originally asked whether the gene expression differences between colon and non-colon samples (as shown in Figure 3b) could result from different isolation protocols. Since our previous response may have been too short, we would like to explain this in more detail. It is known that enzymatic tissue dissociation protocols can have an impact on gene expression programs. In those cases where isolation protocols were compared, e.g. (Denisenko et al., 2020; Mattei et al., 2020; van den Brink et al., 2017), cells were usually treated at 37°C for 1h or longer, and the transcriptional response included the induction of 20-40 genes related to stress response. The work by van den Brink et al (2017) may serve as a showcase for responses in leukocytes upon 1 or 2 hours of enzymatic treatment. Since the enzymatic treatment protocol for colon is significantly shorter (30 min at 37°C only), we may expect even fewer isolation-induced differences in gene expression. Compared to the appr. 2000 genes showing significant differences between colon and non-colon tissue Treg, the number of expected isolation-induced changes appears very small. Hence, we are confident that the observed gene expression differences between colon and non-colon samples (as shown in Figure 3b) do NOT result from different isolation protocols.

To address this issue, we now revised the text to:

“As shown in Fig. 3b, gene expression in colon-derived donor Treg was indeed very different from that of spleen- and liver-derived cells, with many genes significantly upregulated in either colon or spleen/liver (highlighted by coloring in Fig. 3b). **Since only colon cells underwent enzymatic treatment at 37°C during isolation, we cannot definitely exclude a contribution of differential stress responses upon Treg isolation from target tissues to differential gene expression. However, given the observed magnitude of differences between colon and non-colon Treg compared to the few genes known to be induced by enzymatic tissue dissociation procedures (Denisenko et al., 2020; van den Brink et al., 2017), the large majority of changes represent differential tissue imprints.** In line with this, differentially expressed genes overlapped substantially with gene sets (labelled in Fig. 3b) previously classified in a single cell analysis of tissue-resident Treg as non-lymphoid tissue (NLT)- and lymphoid tissue (LT)-specific, respectively”.

7. In the next question the reviewer asked again about a comparison with the blood Tregs. The reviewer raises the question whether the change in the gene profile is due to tissue adaptation or systemic effect. I do not understand this question, as the gene profile is different in the different organs. The authors answered by repeating again that there are differences between the organs and that the analysis of the Tregs in the blood will not add very much. I think I agree with them.

8. The next is about the levels of the IL-2R between the expanded Tregs and the Tregs that were re-isolated. The authors provided a convincing answer in my view.

9. The next question is about stable phenotype and exhaustion. The authors replied convincingly on the stable phenotype and for the exhaustion they provide in the rebuttal letter additional data and I agree with them that the markers that characterize exhausted Tconv are not well characterized for Tregs.

10. The authors have addressed the last two questions about CD62L from the reviewer by adding sentences that clarify their statements.

Reviewer #4 (Remarks to the Author):

All the questions have been fully addressed, so I agree that the revised version can be accepted for publication without further delay.

References:

- Beilhack, A., Schulz, S., Baker, J., Beilhack, G.F., Wieland, C.B., Herman, E.I., Baker, E.M., Cao, Y.-A., Contag, C.H., and Negrin, R.S. (2005). In vivo analyses of early events in acute graft-versus-host disease reveal sequential infiltration of T-cell subsets. *Blood* *106*, 1113-1122.
- Denisenko, E., Guo, B.B., Jones, M., Hou, R., de Kock, L., Lassmann, T., Poppe, D., Clement, O., Simmons, R.K., Lister, R., and Forrest, A.R.R. (2020). Systematic assessment of tissue dissociation and storage biases in single-cell and single-nucleus RNA-seq workflows. *Genome Biol* *21*, 130.
- Mattei, D., Ivanov, A., van Oostrum, M., Pantelyushin, S., Richetto, J., Mueller, F., Beffinger, M., Schellhammer, L., Vom Berg, J., Wollscheid, B., Beule, D., Paolicelli, R.C., and Meyer, U. (2020). Enzymatic Dissociation Induces Transcriptional and Proteotype Bias in Brain Cell Populations. *Int J Mol Sci* *21*.
- van den Brink, S.C., Sage, F., Vertesy, A., Spanjaard, B., Peterson-Maduro, J., Baron, C.S., Robin, C., and van Oudenaarden, A. (2017). Single-cell sequencing reveals dissociation-induced gene expression in tissue subpopulations. *Nat Methods* *14*, 935-936.